# Deterministic nuclear spin squeezing and squeezing by continuous measurement using vector and tensor light shifts

**Ali Moshiri*** **and Alice Sinatra†**

Laboratoire Kastler Brossel, ENS-Université PSL, CNRS, Université de la Sorbonne et Collège de France, 24 rue Lhomond, 75231 Paris, France

⋆ ali.moshiri@lkb.ens.fr , † alice.sinatra@lkb.ens.fr

## Abstract

We study the joint effects of vector and tensor light shifts in a set of large spin atoms, prepared in a polarized state and interacting with light. Depending on the ratio $\epsilon$ between tensor and vector coupling and a measurement rate $\Gamma$, we identify a regime of quantum non-demolition measurement squeezing for times shorter than $(\sqrt{\epsilon}\Gamma)^{-1}$, and a deterministic squeezing regime for times longer than $(\epsilon\Gamma)^{-1}$. We apply our results to fermionic isotopes of strontium, ytterbium, and helium, which are atoms with purely nuclear spin in their ground state, benefiting from very low decoherence. For ytterbium 173, with a cavity such as that of [1], it would be possible to achieve an atomic spin variance reduction of 0.03 in $\simeq$ 50ms.

# 1   Introduction

Atomic sensors based on the precession of a collective spin, sum of all the spins of a set of atoms, reached in the 2000s a precision level close to the standard quantum limit in atomic clocks [2], magnetometers [3, 4], and inertial sensors [5–7]. While these sensors generally use a coherent spin state prepared with independent atoms, spin squeezing, by introducing correlations between atoms [8, 9] and thus allowing to beat the standard quantum limit, saw its first experimental implementations a few years later, mainly with alkali atoms but not exclusively [10–15].

In this article, we are interested in the spin squeezing of alkaline earth atoms such as strontium 87, or similar atoms such as ytterbium 173, some of whose fermionic isotopes have a large purely nuclear spin in the ground state. With long coherence times and narrow optical transitions, these atoms are very useful for optical atomic clocks in the case of strontium [16, 17], while ytterbium 173 could offer interesting prospects for both clocks and magnetometry [18]. Thanks to their large purely nuclear spin, they also represent a promising platform for quantum simulation [19, 20].

Within the theoretical framework developed in this article, we are also interested in nuclear spin squeezing in the ground state of helium-3, which has a spin of 1/2. Due to its exceptional ground state coherence time of over sixty hours [21], this atom is employed in fundamental physics experiments [21, 22] that could benefit from spin squeezing. We propose accessing the nuclear spin of helium-3 via the metastable $2^3S_1$ state of spin 3/2 [23–25]. Another promising method, which involves coupling the collective nuclear spin to a radio frequency circuit, has recently been proposed [26].

To squeeze the collective nuclear spin of these atoms by correlating them with each other, one possible method is to make them interact with light. Under certain conditions, in particular when the light field is highly detuned from an atomic transition that allows an effective Hamiltonian to be derived in the atomic ground state, a quantum non-demolition measure-

ment (QND) of the collective atomic spin fluctuations can be performed using the Faraday effect [27, 28]. This method, which has been extensively tested for alkali atom systems, uses the vector part of the interaction between the atoms and the light field, which introduces a magnetic field-type term into the effective Hamiltonian of the form $F_z S_z$ where $\vec{F}$ denotes the collective atomic spin in the ground state and $\vec{S}$ the Stokes spin describing the degrees of freedom of light polarization. Using the Holstein-Primakoff approximation, this Faraday term can be written $P P_c$ where $P$ and $P_c$ are quadratures of two bosonic modes, for atoms and light respectively. For spin $f > 1/2$ atoms, in addition to this Faraday term, another term taking a simple form $X_A X_L$ appears, reflecting the presence of higher-rank tensors in the atom-light interaction [29–33]. The impact of this tensorial term, breaking the QND character of the interaction, is small in the case of a large detuning with respect to the hyperfine structure of the excited state [34], and can be neutralized by dynamic decoupling [35]. Combined with equal weights, the $P P_c$ and $X X_c$ terms give rise to a two-mode squeezing type interaction entangling light and atoms or to a beam-splitter type interaction useful for quantum memories [31]. With different weights, these two terms can trigger entanglement generated by dissipation as demonstrated with two macroscopic atomic ensembles [36, 37].

In the present work, for a QND-like configuration where the atoms are polarized and the light is linearly polarized in the same direction, the relative weight of the $X X_c$ to the $P P_c$ term in the effective Hamiltonian in the atomic ground state, which we call $\epsilon$, can be tuned by choosing the light frequency. We derive the equations for the system dynamics for any $\epsilon$, and show for $0 < \epsilon \ll 1$ the existence of two spin squeezing regimes: a quasi-QND squeezing regime by continuous homodyne measurement, and a deterministic squeezing regime, absent in the case of spins 1/2. For each regime, we quantify analytically the metrological gain as a function of the atomic parameters, including decoherence. Used directly, the spin squeezed state for helium 3 could be used in magnetometry and improve the accuracy of fundamental physics experiments [22]. As for alkaline earth atoms or similar atoms, the squeezed state could be transferred to an optical transition [38], to benefit atomic clocks [16, 17].

In the following, we derive in section 2 a Hamiltonian that describes the collective behaviour of polarised, large-spin atoms that interact with a detuned, polarised field in a cavity. Section 3 demonstrates the existence of a deterministic spin-squeezing regime using a two-mode master equation, providing numerical estimates for two atomic species: $^{173}$Yb and $^{87}$Sr. We identify the two parameters that govern such a squeezing regime: (i) variance reduction, which scales as $\epsilon$ and depends on the atomic structure, and can be tuned by selecting the atom-light detuning; and (ii) the squeezing rate, which scales as $\epsilon \Gamma$ and depends on the atomic species, as well as the Rabi coupling and the cavity loss rate. Section 4 considers squeezing via continuous homodyne detection of the field leaving the cavity. For small $\epsilon$, we demonstrate the existence of a quasi-QND regime and calculate how the obtained squeezing and squeezing rate scale with respect to $\epsilon$. Lastly, in section 5, we generalise the results of the previous sections so that they can be applied to generating nuclear spin squeezing in $^3$He atoms in their ground state.

## 2   Derivation of a model Hamiltonian

We consider a set of large spin atoms in the ground state interacting with a light field that is highly detuned from the atomic transitions. For a linearly polarized field and atoms polarized in the same direction, we derive in this section a model Hamiltonian that can be reduced to only two bosonic modes, an atomic mode and a light mode, which describe respectively the transverse fluctuations of the collective atomic spin and the Stokes spin of light, orthogonally to the polarization direction. In terms of the two quadratures $X_c, P_c$ of the Stokes spin, which describes the polarization state of light, and the two quadratures $X, P$ resulting from the col-

lective atomic operators, the model Hamiltonian is the sum of two terms (equation (17)). The first Faraday-type term $\Omega_V P P_c$ derived from the vector part of the atom-field interaction allows, by measuring the quadrature $X_c$ of the light, a quantum non-demolition measurement (QND) of $P$ [10, 27, 28]. The second term $\Omega_T X X_c$, derived from the tensor part, which is absent in the case of an atomic spin 1/2 in the ground state, breaks the QND character of the inter-action Hamiltonian. It introduces constraints on spin squeezing by continuous measurement based on the Faraday effect on the one hand, and opens up the possibility of deterministic spin squeezing on the other [36, 37].

## 2.1 Extended Holstein-Primakoff approximation

We consider a cloud of atoms in an electronic ground state $g = nS_j$ with electronic angular momentum $j$ and total angular momentum $f$, polarized in a direction $x$ and interacting with a laser beam also polarized in $x$ and propagating in $z$. If the frequency $\omega$ of the laser is sufficiently detuned from the atomic transitions $nS_j \rightarrow nP_{j'}$, the effective Hamiltonian for a particle in the ground state is written as [32] (see appendix A):

$$h_f = \hbar \alpha^v f_z S_z + \alpha^t \left[ \left( \frac{f(f+1)}{3} - f_z^2 \right) S_0 + (f_x^2 - f_y^2) S_x + (f_x f_y + f_y f_x) S_y \right] \tag{1}$$

where we have introduced the Cartesian components of the total angular momentum of the atom in the ground state $f_x, f_y, f_z$ and those of the Stokes spin of light $S_x, S_y, S_z$:

$$S_x = \frac{1}{2}(a_x^\dagger a_x - a_y^\dagger a_y) \; ; \; S_y = \frac{1}{2}(a_x^\dagger a_y + a_y^\dagger a_x) \; ; \; S_z = \frac{1}{2i}(a_x^\dagger a_y - a_y^\dagger a_x) \; ; \; S_0 = \frac{1}{2}(a_x^\dagger a_x + a_y^\dagger a_y). \tag{2}$$

The operator $2S_0$ represents the total number of photons in the light mode. The constants $\alpha^v$ and $\alpha^t$, whose expressions are given in equation (A.10) of appendix A, represent respectively the coupling of the Stokes spin of light with the vector and tensor components of the atomic spin. They depend on the atomic structure, the detuning between the light frequency and the different atomic transitions, and the Rabi coupling (A.11).

In a dilute sample, the collective Hamiltonian for $n$ atoms $\mathcal{H}_f \equiv \sum_i h_{f,i}$ is written as:

$$\mathcal{H}_f = \hbar \alpha^v F_z S_z + \hbar \alpha^t \left[ n \frac{f(f+1)}{3} S_0 + \Sigma_i f_{x,i}^2 \, S_x + T_{xy} S_y - \frac{1}{2} \Sigma_i \left( f_{z,i}^2 + f_{y,i}^2 \right) a_x^\dagger a_x - \frac{1}{2} \Sigma_i \left( f_{z,i}^2 - f_{y,i}^2 \right) a_y^\dagger a_y \right] \tag{3}$$

where we have rewritten operators $f_z^2 S_0$ and $f_y^2 S_x$ to bring out $a_x^\dagger a_x = S_0 + S_x$ and $a_y^\dagger a_y = S_0 - S_x$, and we have defined:

$$F_z \equiv \Sigma_{i=1}^n f_{z,i} \qquad ; \qquad T_{xy} \equiv \Sigma_{i=1}^n \left( f_{x,i} f_{y,i} + f_{y,i} f_{x,i} \right) \tag{4}$$

In second quantization, we introduce atomic boson operators $a_k^\dagger$ that create a particle in state $|\phi_k\rangle \equiv \left| f, m_f^x = f - k \right\rangle$. In this framework, in the spin state $f$, an atomic collective operator $\mathcal{O} = \sum_{i=1}^n \mathcal{O}^i$, sum of one-particle operators, is written as: $\mathcal{O} = \sum_{k,l=0}^{2f} \langle \phi_l | \mathcal{O} | \phi_k \rangle \, a_l^\dagger a_k$. For an atomic state that remains close to the polarized state $|n : \phi_0\rangle$ with $n$ atoms in $|\phi_0\rangle$, such as the squeezed states of the collective spin that interest us, the matrix elements of $a_l^\dagger a_k$ are of order $n$ for $k = l = 0$, of order $\sqrt{n}$ when $k = 0$ and $l \neq 0$ (or vice versa), and of order 1 otherwise. Similarly, close to the coherent state of light $\left| \alpha_x = \sqrt{n_{ph}} e^{i\varphi_x} \right\rangle$ linearly polarized in the $x$ direction, the matrix elements of $a_x, a_x^\dagger$ are of order $\sqrt{n_{ph}}$ with $n_{ph}$ the number of photons, and those of $a_y, a_y^\dagger$ of order 1.

Assuming $n, n_{ph} \gg 1$, we write $\mathcal{H}_f$ at the dominant order in $n, n_{ph}, \sqrt{n n_{ph}}$. At this order, the

operators $F_z$ and $T_{xy}$ are expressed in terms of two quadratures of a single bosonic mode in the Primakoff approximation [29]:

$$\frac{F_z}{\sqrt{\langle F_x \rangle}} \simeq \frac{F_z}{\sqrt{nf}} \simeq \frac{a_0^\dagger a_1 - a_0 a_1^\dagger}{i\sqrt{2n}} \simeq \frac{a_1 - a_1^\dagger}{i\sqrt{2}} \equiv P \tag{5}$$

$$\frac{T_{xy}}{\sqrt{\langle F_x \rangle}} \simeq (2f-1)\frac{a_0^\dagger a_1 + a_0 a_1^\dagger}{\sqrt{2n}} \simeq (2f-1)\frac{a_1 + a_1^\dagger}{\sqrt{2}} \equiv (2f-1)X \tag{6}$$

Similarly, for Stokes spin components $S_y$ and $S_z$, we can introduce the quadratures of the light mode

$$\frac{S_y}{\sqrt{\langle S_x \rangle}} \simeq \frac{S_y}{\sqrt{n_{\mathrm{ph}}/2}} \simeq \frac{a_x^\dagger a_y + a_x a_y^\dagger}{\sqrt{2n_{\mathrm{ph}}}} \simeq \frac{e^{-i\varphi_x}a_y + e^{i\varphi_x}a_y^\dagger}{\sqrt{2}} \equiv \frac{c + c^\dagger}{\sqrt{2}} \equiv X_c \tag{7}$$

$$\frac{S_z}{\sqrt{\langle S_x \rangle}} \simeq \frac{a_x^\dagger a_y - a_x a_y^\dagger}{i\sqrt{2n_{\mathrm{ph}}}} \simeq \frac{e^{-i\varphi_x}a_y - e^{i\varphi_x}a_y^\dagger}{i\sqrt{2}} \equiv \frac{c - c^\dagger}{i\sqrt{2}} \equiv P_c \tag{8}$$

Let us now examine the remaining atomic operators of the Hamiltonian (3). First, we have:

$$\Sigma_i f_{x,i}^2 = nf^2 + \sum_{k=1}^{2f} k(k-2f)a_k^\dagger a_k \tag{9}$$

On the other hand, since the one-particle operator $f_z^2 + f_y^2$ is none other than $f^2 - f_x^2$, we have:

$$\Sigma_i \left( f_{z,i}^2 + f_{y,i}^2 \right) = nf(f+1) - \left( nf^2 + \sum_{k=1}^{2f} k(k-2f)a_k^\dagger a_k \right) \tag{10}$$

As for $\Sigma_i \left( f_{z,i}^2 - f_{y,i}^2 \right)$, its matrix elements are of order $\sqrt{n}$, so $\left( \Sigma_i \left( f_{z,i}^2 - f_{y,i}^2 \right) \right) \otimes a_y^\dagger a_y$ is of order $\sqrt{n}$. Limiting ourselves to order $n, n_{\mathrm{ph}}, \sqrt{nn_{\mathrm{ph}}}$, this is therefore a negligible term. By expanding $S_0$, using expressions (9) and (10), and grouping what is in $a_x^\dagger a_x$ on the one hand and $a_y^\dagger a_y$ on the other, the Hamiltonian (3) can be rewritten as follows:

$$\mathcal{H}_f = \hbar \alpha^v F_z S_z + \hbar \alpha^t T_{xy} S_y + \hbar \alpha^t \left[ a_x^\dagger a_x \left( \frac{nf}{3}(2f-1) + \sum_{k=1}^{2f} k(k-2f)a_k^\dagger a_k \right) - a_y^\dagger a_y \frac{nf}{3}\left(f - \frac{1}{2}\right) \right] \tag{11}$$

To the atom-light interaction (11), we add the cavity Hamiltonian $\mathcal{H}_c = \hbar\omega_c(a_x^\dagger a_x + a_y^\dagger a_y)$, a Hamiltonian $\mathcal{H}_L = i\hbar(\beta e^{-i\omega t}a_x^\dagger - \beta^* e^{i\omega t}a_x)$ representing a coherent field at angular frequency $\omega$ injected into the cavity, polarized along $x$ and with amplitude $\beta$, as well as a static magnetic field along $x$, $\mathcal{H}_B = -\hbar\gamma_f B_0 F_x = -\hbar\gamma_f B_0\left(nf - \sum_{k=1}^{2f} ka_k^\dagger a_k\right)$, where $\gamma_f$ is the gyromagnetic ratio of the spin atoms $f$, $B_0$ is the scalar value of the magnetic field, and $F_x$ is the collective atomic angular momentum operator along $x$. The total Hamiltonian $\mathcal{H}$ is written as:

$$\mathcal{H} = \mathcal{H}_f + \mathcal{H}_c + \mathcal{H}_L + \mathcal{H}_B \tag{12}$$

In the rotating frame $\tilde{a}_{x,y} = a_{x,y}e^{i\omega t}$, and at order $n, n_{\mathrm{ph}}, \sqrt{nn_{\mathrm{ph}}}$, we can write, from expressions (5), (6), (11) and (12), the linearized equations of motion and find the Hamiltonian that describes the transverse degrees of freedom of our atom-light system $\mathcal{H}$ (see Appendix B):

$$\mathcal{H} = \hbar\Omega_V P P_c + \hbar\Omega_T X X_c + \hbar\left(\tilde{\delta} - \alpha^t nf(f-1/2)\right)\frac{X_c^2 + P_c^2}{2} + \hbar\delta_B(1)\frac{X^2 + P^2}{2} + \hbar\sum_{k=2}^{2f}\delta_B(k)\frac{X_k^2 + P_k^2}{2} \tag{13}$$

where we have introduced:

$$\tilde{\delta} = \delta_c + \alpha^t \frac{nf}{3}(2f-1)$$
$$\delta_B(k) = k\gamma_f B_0 - \alpha^t n_{\mathrm{ph}}k(2f-k) \tag{14}$$

with $\delta_c = \omega_c - \omega$ the empty cavity detuning. $\tilde{\delta}$ represents the cavity detuning in the presence of large spin atoms for the polarized field $x$ that is injected into the cavity (see (B.2)); as for $\delta_B(k)$, it is the sum of a Zeeman shift and a light shift. In the first two terms of (13), we have introduced the vector coupling constant $\Omega_V$ and the tensor coupling constant $\Omega_T$:

$$\Omega_V \equiv \sqrt{\frac{n\, n_{\mathrm{ph}}f}{2}}\alpha^v \quad ; \quad \Omega_T \equiv \sqrt{\frac{n\, n_{\mathrm{ph}}f}{2}}(2f-1)\alpha^t \tag{15}$$

We also introduce the ratio of the two couplings, which will be decisive in the following:

$$\epsilon = \frac{\Omega_T}{\Omega_V}. \tag{16}$$

This parameter $\epsilon$, determined by the atomic structure and atom-light detuning, can be positive or negative. As we show however in section 3, squeezing only exists in the case $\epsilon > 0$. In the first two terms of the Hamiltonian (13), we recognize a term of vector origin describing the Faraday effect, to which is added a second term of tensor origin that does not commute with the first. The tensor term $XX_c$ comes from the operator $T_{xy}S_y$. In other words, near the polarized state, the atomic collective operator corresponding to the one-particle operator $f_x f_y + f_x f_y$ acts on the same atomic mode as $F_z$. Also, by carefully choosing the cavity detuning and the magnetic field to obtain $\tilde{\delta} = \alpha^t nf(f-1/2)$, $\delta_B(1) = 0$, the final model Hamiltonian can be summarized by the first two terms of (13):

$$\mathcal{H} = \hbar\Omega_V PP_c + \hbar\Omega_T XX_c \tag{17}$$

Thus, only the first atomic bosonic mode couples to the $y$ mode in the cavity. The other atomic modes are decoupled from the light.[1]

## 2.2 Two-mode master equation

The density operator $\rho$ of the system evolves according to a master equation with the Hamiltonian (17) to which Lindblad terms describing photon losses in the cavity with a rate $\kappa$ must be added. We also consider a possible rate $\gamma$ decoherence for the atomic mode which may arise, for example, from spontaneous emission [19] and optical pumping required to maintain the atoms in the fully polarized state.

$$\frac{d\rho}{dt} = \frac{1}{i\hbar}[\hbar\Omega_V(PP_c + \epsilon XX_c), \rho] + \kappa\left(c\rho c^\dagger - \frac{1}{2}\{c^\dagger c, \rho\}\right) + \gamma\left(a\rho a^\dagger - \frac{1}{2}\{a^\dagger a, \rho\}\right) \tag{18}$$

The annihilation operators of a bosonic excitation in the atomic mode $a \equiv a_1$ and photonic mode $c \equiv e^{-i\varphi_x}a_y$ were introduced in (5) and (7). Initially, with the system in the completely polarized state $|n : \phi_0\rangle \otimes |\alpha_x\rangle$, the atomic mode and the photonic mode are in the vacuum of bosonic excitations. Here, we are interested in the case $\kappa \gg \Omega_V, \gamma$, for which the mode $X_c$, $P_c$ of the electromagnetic field in the cavity rapidly evolves towards a steady state adapting to the slow evolution of the mode $X, P$ of the atomic spin. We also assume $0 < \epsilon < 1$. In this context, we first study in section 3 a deterministic spin squeezing scheme made possible by the presence of the tensor term, then we quantify in section 4 the influence of the tensor term on spin squeezing by continuous homodyne measurement of the field leaving the cavity.

---

[1]Note that compared to [39], our model Hamiltonian couples only the first atomic mode to light because we take a polarized state as the atomic reference state.

## 3   Deterministic spin squeezing

### 3.1   Deterministic squeezing from the two-mode master equation

In this section, we integrate the equations of motion of the second-order moments of the quadratures, describing the coupled fluctuations of the collective atomic spin and the Stokes spin. With the two-mode master equation (18), we obtain two closed systems of equations, each involving a single atomic quadrature $X$ or $P$:

$$
\begin{aligned}
\frac{d}{dt}\langle P^2\rangle &= -2\Omega_T\langle PX_c\rangle - \gamma\left(\langle P^2\rangle - \frac{1}{2}\right) & \frac{d}{dt}\langle X^2\rangle &= 2\Omega_V\langle XP_c\rangle - \gamma\left(\langle X^2\rangle - \frac{1}{2}\right)\\
\frac{d}{dt}\langle PX_c\rangle &= -\frac{\kappa+\gamma}{2}\langle PX_c\rangle + \Omega_V\langle P^2\rangle - \Omega_T\langle X_c^2\rangle & \frac{d}{dt}\langle XP_c\rangle &= -\frac{\kappa+\gamma}{2}\langle XP_c\rangle - \Omega_T\langle X^2\rangle + \Omega_V\langle P_c^2\rangle\\
\frac{d}{dt}\langle X_c^2\rangle &= -\kappa\left(\langle X_c^2\rangle - \frac{1}{2}\right) + 2\Omega_V\langle PX_c\rangle & \frac{d}{dt}\langle P_c^2\rangle &= -\kappa\left(\langle P_c^2\rangle - \frac{1}{2}\right) - 2\Omega_T\langle XP_c\rangle
\end{aligned}
$$

(19)

Before solving this system exactly, let us look at how, in case $\gamma = 0$, $\langle PX_c\rangle$ and $\langle P^2\rangle$ couple in the system on the left. In the steady state:

$$
\begin{aligned}
\langle PX_c\rangle &= \frac{2\Omega_V}{\kappa}\left(\langle P^2\rangle - \frac{\epsilon}{2}\right) - \frac{2\Omega_T}{\kappa}\left(\langle X_c^2\rangle - \frac{1}{2}\right)\\
\langle X_c^2\rangle - \frac{1}{2} &= \frac{2\Omega_V}{\kappa}\langle PX_c\rangle
\end{aligned}
$$

(20)

which shows that $\langle PX_c\rangle$ replicates $\langle P^2\rangle - \epsilon/2$ with a rate of $2\Omega_V/\kappa$. Since $\langle P^2\rangle - \epsilon/2$ couples with $\langle PX_c\rangle$ at a rate $-2\Omega_T$ according to the first equation on the left part of the system (19), the evolution of $\langle P^2\rangle - \epsilon/2$ is ultimately damped with a characteristic coefficient $-\frac{4\Omega_T\Omega_V}{\kappa}$. For the quadrature $P$ to be squeezed, $\Omega_T$ and $\Omega_V$ must therefore have the same sign with $|\Omega_T| < |\Omega_V|$, i.e. $0 < \epsilon < 1$. Taking into account decoherence, we obtain the differential equations verified respectively by $\langle P^2\rangle$ and $\langle X^2\rangle$, by adiabatic elimination of the fast components:

$$
\begin{aligned}
\frac{d}{d\tau}\langle P^2\rangle &= -\frac{2\epsilon}{1 + \frac{\tilde\gamma}{\tilde\kappa} + \frac{2\epsilon}{\tilde\kappa}}\left(\langle P^2\rangle - \frac{\epsilon}{2}\right) - \tilde\gamma\left(\langle P^2\rangle - \frac{1}{2}\right)\\
\frac{d}{d\tau}\langle X^2\rangle &= -\frac{2\epsilon}{1 + \frac{\tilde\gamma}{\tilde\kappa} + \frac{2\epsilon}{\tilde\kappa}}\left(\langle X^2\rangle - \frac{1}{2\epsilon}\right) - \tilde\gamma\left(\langle X^2\rangle - \frac{1}{2}\right)
\end{aligned}
$$

(21)

We have introduced $\tau$, the dimensionless time, by the rate $\Gamma$, and normalized $\gamma, \kappa$ by $\Gamma$:

$$
\tau \equiv \Gamma t \quad ; \quad \Gamma \equiv \frac{2\Omega_V^2}{\kappa} \quad ; \quad \tilde\gamma = \frac{\gamma}{\Gamma} \quad ; \quad \tilde\kappa = \frac{\kappa}{\Gamma}
$$

(22)

Note that $\tilde\gamma^{-1} = \frac{2\Omega_V^2}{\kappa\gamma}$ corresponds to the cooperativity $C$ of the coupled atom-field system. In the limit $\tilde\gamma, \epsilon \ll \tilde\kappa$, the solution of (21) for $\langle P^2\rangle$ is written as:

$$
\langle P^2\rangle = \frac{1}{2}\frac{\epsilon + \tilde\gamma/2\epsilon}{1 + \tilde\gamma/2\epsilon} + \frac{1}{2}\frac{1-\epsilon}{1 + \tilde\gamma/2\epsilon}e^{-(2\epsilon+\tilde\gamma)\tau} \underset{\gamma\ll\epsilon\Gamma}{\simeq} \frac{\epsilon}{2} + \frac{1-\epsilon}{2}e^{-2\epsilon\tau}
$$

(23)

In the deterministic regime, made possible by the presence of the tensor term ($\epsilon \neq 0$), and for $\gamma \ll \epsilon\Gamma$, the squeezing therefore occurs at a rate of $2\epsilon\Gamma$, with an asymptotic noise reduction equal to $\epsilon$: [2]

$$
\tau_{\text{deter}} \simeq \frac{1}{2\epsilon} \; ; \; \langle P\rangle_{\text{deter}} = 0 \; ; \; \Delta P_{\text{deter}}^2 \simeq \frac{\epsilon}{2} \; ; \; \langle X\rangle_{\text{deter}} = 0 \; ; \; \Delta X_{\text{deter}}^2 \simeq \frac{1}{2\epsilon}
$$

(24)

It is therefore sufficient to have either $0 < \epsilon < 1$ and the squeezing is done in $P$, or else $\epsilon > 1$

---

[2]Expressions (23) and (24) clearly show the need for $\epsilon > 0$. Another, perhaps more intuitive manner to show the role of $\epsilon$ and the need to have $\epsilon > 0$, using Langevin equations, is presented in appendix C.

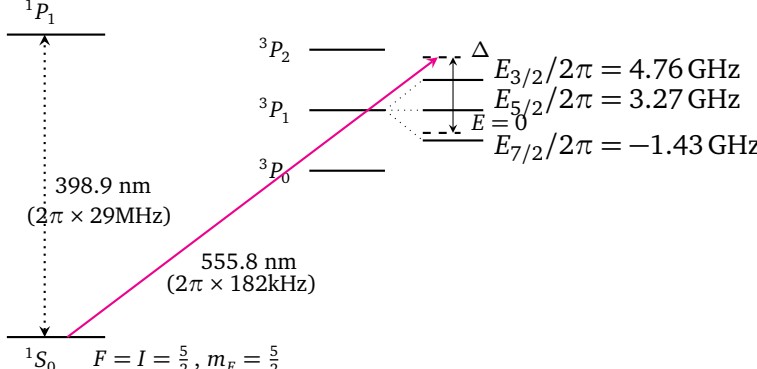

Figure 1: Diagram of the $^1S_0 \to {}^3P_1$ transition for $^{173}$Yb. $\Delta$ is the detuning between the light and the $^1S_0 \to {}^3P_0$ transition of $^{176}$Yb. Hyperfine level energy values extracted are taken from [40].

and then $X$ is squeezed. Finally, note that our calculation close to the polarized state remains valid as long as $a_1^\dagger a_1 \ll a_0^\dagger a_0$, i.e. $\frac{1}{2\epsilon} \ll n$.

## 3.2 Application to ytterbium 173

In this section, we apply the analytical results seen previously to ytterbium $^{173}$Yb, which has a transition $^1S_0 \to {}^3P_1$ (Figure 1). The vector and tensor coupling constants (A.10) as a function of the detuning are plotted in Figure (2). From this plot, we see that in order to have $\epsilon$ small, i.e., the best possible squeezing, while keeping $\epsilon > 0$, we have to be on the right of the graph, in other words as in Figure (1), above the energy level $E_{3/2}$. More generally, for a very large detuning, both the vector $\alpha^v$ and tensor $\alpha^t$ couplings decrease, the tensor coupling decreasing faster than the vector one (see e.g. [19,32]). Since the theoretical squeezing limit is $\epsilon = \alpha^t/\alpha^v$, the larger the detuning, the better the squeezing. However, as the squeezing rate is $\epsilon\Gamma \propto \alpha^v\alpha^t$, the larger the detuning, the slower the squeezing.. In a cavity like in [16], Figure (3) shows that for a detuning of 9GHz, i.e. close to the cancellation point of $\alpha^t$, a reduction in the variance of P of $\epsilon = 0.03$ is obtained with a deterministic squeezing rate $2\epsilon\Gamma$ of $44.7\,\mathrm{s}^{-1}$. Such a favourable combination of squeezing level and squeezing rate stems from the existence of a detuning window where $\alpha^t$ tends to zero while $\alpha^v$ (hence $\Gamma$) is relatively large. At 9.6GHz, it would even theoretically be possible to achieve a reduction in the variance of P of 0.005 for a squeezing rate of $2\epsilon\Gamma \simeq 2.8\,\mathrm{s}^{-1}$.

## 3.3 Application to strontium 87

In the same way as for ytterbium in the previous section, we apply our analytical results to the fermionic isotope of strontium, $^{87}$Sr, which in its ground state has a purely nuclear spin $I = 9/2$ and is used in optical transition clocks [16,17]. In Figures 4 and 5, we have plotted the reduction in variance of $P$ and the deterministic squeezing rate as a function of the frequency detuning, with respect to the energy level $^3P_1$ of the fine structure of strontium [19]. With a cavity of type [1], and for a detuning of 3 GHz, the deterministic squeezing time is of the order of the second for an asymptotic variance reduction of P of $\epsilon = 0.35$ (fig. 4). A cavity with a strong Rabi coupling [41], and being even more detuned $\simeq 10$GHz, would allow a greater metrological gain ($\epsilon \simeq 0.08$), in a reasonable squeezing rate of $\simeq 600\,\mathrm{s}^{-1}$ (fig. 5).

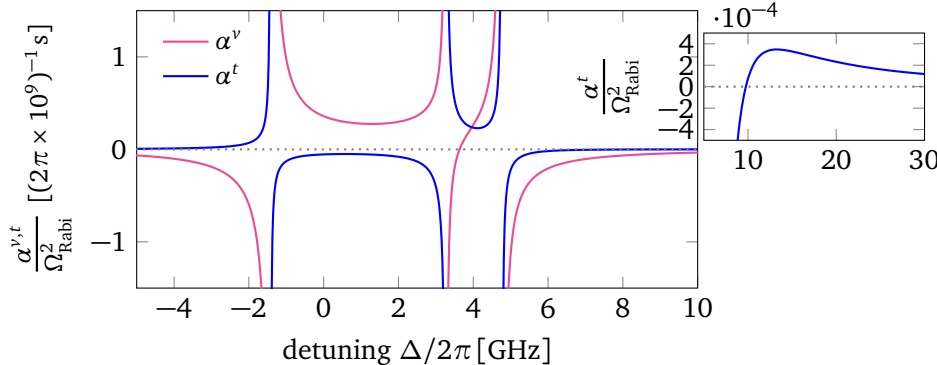

Figure 2: Vectorial and tensorial coupling constants (A.10), divided by $\Omega_{\text{Rabi}}^2$ (A.11), in units of $(2\pi \times 10^9)^{-1}$ s, for the polarized state $F = 5/2$, i.e. on the $^1S_0 \to {}^3P_1$ transition, as a function of the frequency detuning in GHz for $^{173}$Yb, with respect to the $^1S_0 \to {}^3P_0$ transition of $^{176}$Yb. Spontaneous emission is neglected here and the coupling constants are therefore real. Top right: zoom on the tensor coupling by a factor $10^4$ for the detuning window where it vanishes .

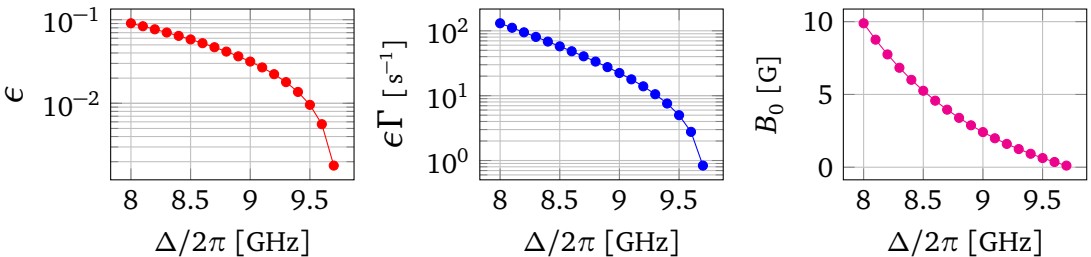

Figure 3: $^{173}$Yb. From left to right, as a function of the detuning (in GHz): (left) $\epsilon = \Delta P_{\text{deter}}^2/\Delta P_{t=0}^2$ inverse of the metrological gain. (middle) $\epsilon\Gamma$ deterministic squeezing rate divided by 2. (right) $B_0$ magnetic field along $x$ to compensate for the lightshift. Cavity parameters [1]: $\kappa = 2\pi \times 153$kHz; $n_{\text{ph}} = 7.3 \times 10^5$; $\Omega_{\text{Rabi}} = 2\pi \times 21.7$kHz; $n_{\text{at}} = 2 \times 10^4$.

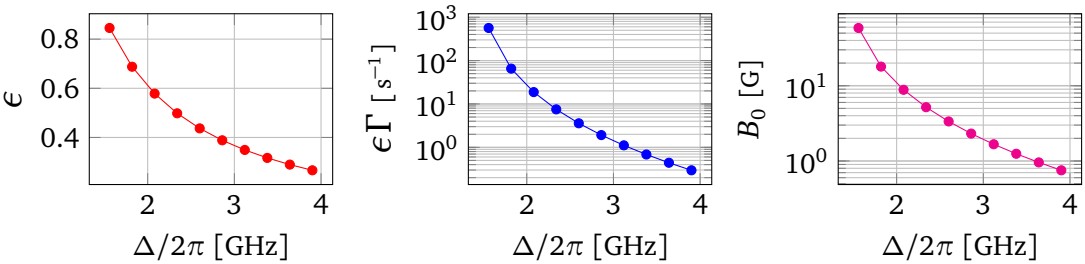

Figure 4: $^{87}$Sr. From left to right, depending on the detuning (in GHz): (left) $\epsilon = \Delta P_{\text{deter}}^2/\Delta P_{t=0}^2$ inverse of the metrological gain. (middle) $\epsilon\Gamma$ deterministic squeezing rate divided by 2. (right) $B_0$ magnetic field along $x$ to compensate for lightshift. Cavity parameters [1]: $\kappa = 2\pi \times 153$kHz; $n_{\text{ph}} = 9 \times 10^5$; $\Omega_{\text{Rabi}} = 2\pi \times 5.5$kHz; $n_{\text{at}} = 2 \times 10^4$.

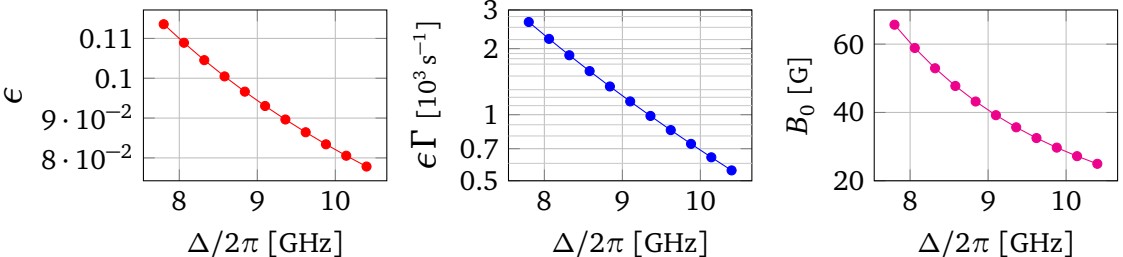

Figure 5: $^{87}$Sr. From left to right, depending on the detuning (in GHz): (left) $\epsilon = \Delta P_{\mathrm{deter}}^2 / \Delta P_{t=0}^2$ inverse of metrological gain. (middle) $\epsilon \Gamma$ deterministic squeezing rate divided by 2. (right) $B_0$ magnetic field along $x$ to compensate for lightshift. Cavity parameters [41]: $\kappa = 2\pi \times 10\mathrm{MHz}$; $n_{\mathrm{ph}} = 2.8 \times 10^5$; $\Omega_{\mathrm{Rabi}} = 2\pi \times 937\mathrm{kHz}$; $n_{\mathrm{at}} = 2 \times 10^4$.

## 4  Squeezing by continuous homodyne detection

Here we are interested in the evolution of the atomic state conditioned to the result of a continuous measurement of the field leaving the cavity, as shown in Figure 6. Following the

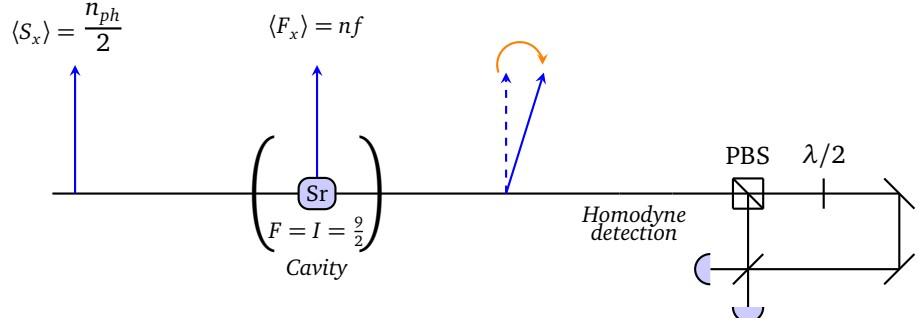

Figure 6: Schematic diagram of spin squeezing by Faraday effect and continuous homodyne measurement of the field leaving the cavity. A cloud of atoms in the ground state and polarized along the $x$ axis is placed in an optical cavity with axis $z$ into which a coherent field polarized in the same direction $x$ is injected. By the Faraday effect, the atoms induce a small rotation of the polarization proportional to the component $F_z$ of the collective atomic spin, which can therefore be measured indirectly by performing homodyne detection of the outgoing field polarized along $y$. PBS: polarizing beam splitter.

procedure in [23], we introduce the integrated homodyne signal

$$\sigma(t) = \frac{N_+^{\mathrm{tot}} - N_-^{\mathrm{tot}}}{2\mu t} \tag{25}$$

proportional to the difference in the number of photons recorded in the two channels in the time interval between 0 and $t$, where $\mu^2$ has the dimension of an angular frequency, and we will calculate the mean and variance of the atomic quadrature $P$, conditioned to a measurement result $\sigma = \mathcal{S}$ for the signal. For $\epsilon = 0$, we know from reference [23] that, during continuous measurement, the atoms evolve towards a spin-squeezed state with a mean value of $P$ proportional to $\mathcal{S}$, and a conditional variance of $P$ increasingly reduced relative to the standard quantum limit. Here we quantify the influence of the tensor term ($\epsilon \neq 0$), which breaks the quantum non-demolition character of the interaction.

### 4.1 Derivation of a master equation for the atomic mode

To simplify the presentation and extract the physics introduced by the tensor term, we first consider the case where the atomic mode is undamped, $\gamma = 0$. Since the cavity mode is strongly damped, we can eliminate it adiabatically to obtain a new single-mode master equation describing the slow evolution of the atomic operators. We perform the adiabatic elimination using the Monte Carlo wave function formalism [42], where the density operator solution of the master equation (18) is obtained by averaging pure states over independent stochastic realizations, each realization corresponding to the deterministic evolution of an unnormalized state vector $|\psi(t)\rangle$ under the action of the non-Hermitian effective Hamiltonian:

$$H_{\text{eff}} = \hbar\Omega_V(PP_c + \epsilon XX_c) - \frac{i\hbar}{2}\kappa c^\dagger c \,, \tag{26}$$

randomly interrupted by quantum jumps of the jump operator $C_c = \sqrt{\kappa}c$. In the absence of coupling (i.e., for $\Omega_V = 0$), the atomic mode and the cavity mode remain in their initial state, i.e., the vacuum state. In the first order in $\Omega_V$, i.e. by a single action of the effective Hamiltonian, this state is coupled to states with one excitation in the cavity mode by the action of $P_c$ and $X_c$. As we showed in [23], in the weak coupling limit, i.e. for $\Omega_V/\kappa \to 0$, we can truncate the Monte Carlo state vector in the Fock basis of the cavity $|\psi\rangle = \sum_{n_c} |\psi^{n_c}\rangle_{\text{at}} |n_c\rangle_{\text{cav}}$ at first order in $\Omega_V$:

$$|\psi\rangle = |\psi^0\rangle|0\rangle + |\psi^1\rangle|1\rangle \tag{27}$$

Looking at the evolution of $|\psi\rangle$ under the effect of the effective Hamiltonian (26), and projecting onto $|1\rangle$, we find that the fast component $|\psi^1\rangle$ exponentially reaches an adiabatic following regime of the slow component $|\psi^0\rangle$, hence its adiabatic elimination:

$$|\psi^1\rangle \approx \frac{\sqrt{2}}{\kappa}(\Omega_V P - i\Omega_T X)|\psi^0\rangle \tag{28}$$

expression that we can replace into the evolution equation of $|\psi^0\rangle$ obtained by projecting onto $|0\rangle$:

$$i\hbar\frac{d|\psi^0\rangle}{dt} = -i\hbar\frac{\Omega_V^2}{\kappa}(P + i\epsilon X)(P - i\epsilon X)|\psi^0\rangle \equiv -\frac{i\hbar}{2}\Gamma\,C^\dagger C\,|\psi^0\rangle \equiv H_{\text{eff}}^0|\psi^0\rangle \tag{29}$$

We thus obtain the single-mode master equation that describes the slow evolution of the density operator of the undamped mode under the influence of the Hamiltonian $H_{\text{eff}}^0$ and the associated quantum jumps:

$$\frac{d\rho^0}{dt} = \Gamma\left(C\rho^0 C^\dagger - \frac{1}{2}\left\{C^\dagger C, \rho^0\right\}\right) \tag{30}$$

It involves the rate $\Gamma$ already introduced in section 3 eq. (22), as well as the non-Hermitian jump operator

$$C \equiv P - i\epsilon X \tag{31}$$

The non-Hermitian term $-i\epsilon X$, absent in Faraday's purely vectorial Hamiltonian, breaks the QND character of the measurement.

### 4.2 Evolution conditioned to a continuous measurement

#### 4.2.1 Evolution for one single realization

In order to describe the evolution of the system conditioned to a continuous measurement by homodyne detection of the field at the cavity output, we reformulate the master equation (30)

in terms of the stochastic evolution of pure states, over which we must average to obtain the mean values given by the master equation. Each pure state evolves according to a continuous-time stochastic equation in the Ito sense[3] [23, 43–45]:

$$d\,|\phi\rangle = -\Gamma\frac{dt}{2}\left(C^\dagger C - 2\bar{p}C + \bar{p}^2\right)|\phi\rangle + \sqrt{\Gamma}d\zeta_s\,(C-\bar{p})|\phi\rangle \quad \text{with} \quad \bar{p} = \langle\phi(t)|\,P\,|\phi(t)\rangle \tag{32}$$

where, for each (non-Hermitian) jump operator $C$ of the master equation, we associate a continuous-time stochastic process $d\zeta_s(t)$ with real values, Gaussian, with zero mean, variance dt, and without memory. Physically, $d\zeta_s(t)$ is the noise of the homodyne signal (eq. (56) in [23]). Equation (32) can be solved exactly using a Gaussian ansatz for the wave function [46] in momentum space, real and normalized to unity:

$$\phi(p,t) = e^{-S} \quad \text{with} \quad S = u(t)(p-\bar{p}(t))^2 - W(t) \tag{33}$$

where $W$ represents the normalization factor. With this ansatz:

$$\frac{d\phi}{\phi} = 2ud\bar{p}\,(p-\bar{p}) + \left(-du + 2u^2d\bar{p}^2\right)(p-\bar{p})^2 - ud\bar{p}^2 + dW \tag{34}$$

Furthermore, by injecting the ansatz (33) into equation (32):

$$\frac{d\phi}{\phi} = -\Gamma\frac{dt}{2}\Big\{(1-4u^2\epsilon^2)(p-\bar{p})^2 + 4\bar{p}u\epsilon\,(p-\bar{p}) +$$
$$2u\epsilon^2 - \epsilon\Big\} + \sqrt{\Gamma}d\zeta_s\,(1-2u\epsilon)(p-\bar{p}) \tag{35}$$

By identifying the terms in $(p-\bar{p})$, $(p-\bar{p})^2$, after calculation, we obtain the differential equations verified by $u(t)$ and $\bar{p}(t)$:

$$du = dt\,\Gamma(1-2u\epsilon)$$
$$d\bar{p} = -\epsilon\Gamma\bar{p}\,dt + d\zeta_s\sqrt{\Gamma}\left(\frac{1}{2u} - \epsilon\right) \tag{36}$$

with the initial conditions $u(0) = \frac{1}{2}$ and $\bar{p} = 0$. Note that, for $\epsilon \neq 0$, $u(\tau) \underset{\tau\gg 1}{\longrightarrow} \frac{1}{2\epsilon}$ and therefore in the differential equation (36) verified by $\bar{p}$, the function $\left(\frac{1}{2u} - \epsilon\right)$ which is in front of the noise tends towards 0 for long times compared to $\frac{1}{\epsilon\Gamma}$ regardless of the trajectory, which also points to the existence of a deterministic spin squeezing regime. The system (36) can be solved analytically:

$$u(\tau) = \frac{1}{2\epsilon}\Big(1-(1-\epsilon)e^{-2\epsilon\tau}\Big) \tag{37}$$

$$\bar{p}(\tau) = e^{-\epsilon\tau}\int_0^\tau e^{\epsilon\tau'}w(\tau',d\zeta_s(\tau')) \tag{38}$$

where we have set $\tau \equiv \Gamma t$ and $w(\tau,d\zeta_s) \equiv d\zeta_s(\tau)\left(\frac{1}{2u(\tau)} - \epsilon\right)$. In Figures 7 and 8, we have plotted, for four realizations of the experiment, the time evolution of $\bar{p}$, the quantum average value of the quadrature $P$ in state $|\phi\rangle$, as well as the variance of $P$, independent of the trajectory, in two situations: in the absence of a tensor term ($\epsilon = 0$, Fig. 7) and in the presence

---

[3]Equation (32) is equivalent to the stochastic master equation:

$$d\rho = \Gamma dt\left[C\rho C^\dagger - \frac{1}{2}\left\{C^\dagger C, \rho\right\}\right] + \sqrt{\Gamma}d\zeta_s\left[(C-\bar{p})\rho + \rho(C^\dagger-\bar{p})\right] \quad ; \quad \rho = |\phi\rangle\langle\phi|$$

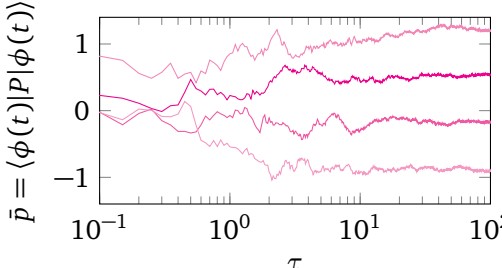 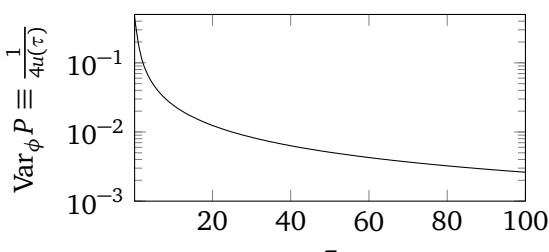

Figure 7: QND measurement of quadrature $P$ by continuous measurement of $X_c$ in the absence of a tensor term: on the left, quantum average value of $P$ for four realizations of the experiment: each trajectory converges to a fixed but unpredictable value; on the right, variance of $P$ (see (D.8)), independent of the trajectory. Parameters: $\epsilon = 0$; $\tilde{\gamma} = 10^{-3}$.

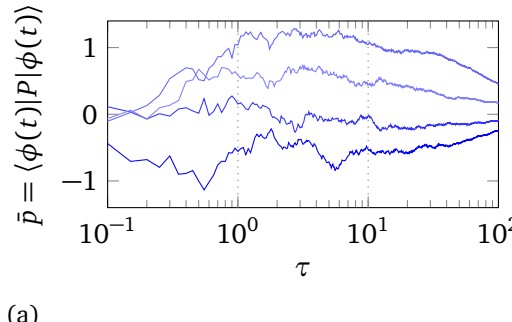 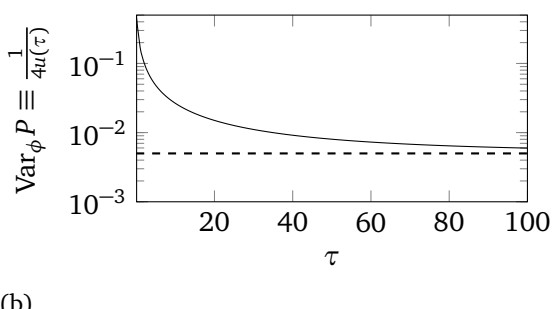

(a)                                                    (b)

Figure 8: Quasi-QND measurement of quadrature $P$ by continuous measurement of $X_c$ in the presence of the tensor term: on the left, quantum mean value of $P$ for four realizations of the experiment: existence of a time window that we refer to as quasi-QND (between the two vertical dotted lines). On the right, the variance of $P$ (see (D.8)), independent of the trajectory, remains greater than $\epsilon/2$ (dashed line). Parameters: $\epsilon = 10^{-2}$; $\tilde{\gamma} = 10^{-3}$.

of a tensor term ($\epsilon \neq 0$, Fig. 8). In case $\epsilon = 0$, $\bar{p}$ converges to a fixed but unpredictable value, reflecting the QND nature of the measurement, and the variance tends towards zero in the absence of atomic decoherence, since $u(t) = 1/2 + \Gamma t$, and towards a small value $\sim \tilde{\gamma}/4$ (see (D.8)) for $\tilde{\gamma} \neq 0$. In the presence of the tensor term, we can notice that for long times, in fact of the order of $\frac{1}{2\epsilon}$ according to the previous section, we can see the onset of deterministic squeezing on each realization of the experiment: regardless of the stochastic trajectory, $\bar{p}$ converges to 0. We also see that there is a time window during which $\bar{p}$ tends to stabilize at a random value (Fig. 8a), with decreasing fluctuations (Fig. 8b), a regime that we will refer to as quasi-QND.

### 4.2.2 Evolution conditioned to the integrated homodyne signal

The mean and variance of an observable in a single realization of the experiment generally have no physical meaning. In practice, rather than the homodyne history, i.e. the detailed time dependence of the homodyne detection signal, we are interested in its temporal mean $\sigma$ defined in (25) over a time interval $[0, t]$ that is easily accessible in an experiment, especially since the presence of decoherence adds atomic noise that is not measured. We will therefore

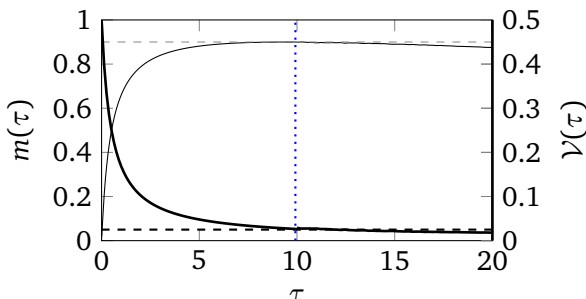

Figure 9: Quasi-QND measurement of $P$ by continuous measurement of $X_c$: temporal evolution of the mean $m(\tau)$ (thin line) and variance $\mathcal{V}(\tau)$ of $P$ (thick line) conditioned to the signal. The blue dotted line shows the squeezing time in $1/\sqrt{\epsilon}$ corresponding to the maximum signal. For this time, the variance conditioned to the signal is $\mathcal{V}_{\text{QND}} \simeq \frac{\sqrt{\epsilon}}{4}$ (thick black dotted line). Parameters: $\epsilon = 10^{-2}$; $\tilde{\gamma} = 10^{-3}$.

focus on the mean and variance of the nuclear spin quadrature $P$ conditioned to the integrated signal $\sigma$, which is rewritten in the stochastic reformulation [23]:

$$\sigma(t) = \frac{1}{t} \int_0^t dt' \left( \sqrt{\frac{\kappa}{2}} \left\langle \phi(t') \right| X_c \left| \phi(t') \right\rangle + \frac{1}{2} \frac{d\zeta_s(t')}{dt'} \right). \tag{39}$$

Remarkably, despite the presence of the tensor term, we can relate the integrated signal (39) to $\bar{p}$ using expressions (27) and (28) of the wave function in the truncated basis (see Appendix D). Thanks to the Gaussian nature of the probability distributions of $\bar{p}$ and $\sigma$, it is then possible to show that the conditional mean is always proportional to the signal, and that the conditional variance, the inverse of the metrological gain, depends on time but not on the signal [23]:

$$\langle P \rangle_{\sigma=\mathcal{S}} = m(t) \frac{\mathcal{S}}{\sqrt{\Gamma}} \quad ; \quad \text{Var}_{\sigma=\mathcal{S}}(P) = \mathcal{V}(t) \tag{40}$$

In Figure 9, we have plotted the mean $m(t)$ and the variance $\mathcal{V}(t)$, conditioned to the integrated signal. Their calculation, including the effect of decoherence at rate $\gamma$, is detailed in Appendix D. We find that for $\epsilon \neq 0$, $m(\tau)$ reaches a maximum. We consider this maximum as the upper time bound $\tau_{\text{QND}}^{\text{max}}$ of the quasi-QND window. We give the results directly here for $\epsilon, \tilde{\gamma} \ll 1$ [4]:

$$\tau_{\text{QND}}^{\text{max}} \simeq \frac{1}{\sqrt{\epsilon + \frac{\tilde{\gamma}}{6}}} \; ; \; m_{\text{QND}} \simeq 1 - \sqrt{\epsilon + \frac{\tilde{\gamma}}{6}} \; ; \; \mathcal{V}_{\text{QND}} \simeq \frac{1}{4} \sqrt{\epsilon + \tilde{\gamma}/6} + \frac{1}{6} \frac{\tilde{\gamma}}{\sqrt{\epsilon + \tilde{\gamma}/6}} \tag{41}$$

At the limit $\epsilon \to 0$, this gives back the scaling law with exponent $-1/2$, which is usual in alkalis and links the optimal spin variance to the cooperativity $\tilde{\gamma}^{-1}$ [47]. In the regime $\tilde{\gamma} \ll \epsilon$, the expressions (41) are simplified:

$$\tau_{\text{QND}}^{\text{max}} \underset{\tilde{\gamma} \ll \epsilon}{\simeq} \frac{1}{\sqrt{\epsilon}} \; ; \; m_{\text{QND}} \underset{\tilde{\gamma} \ll \epsilon}{\simeq} 1 - \sqrt{\epsilon} \; ; \; \mathcal{V}_{\text{QND}} \underset{\tilde{\gamma} \ll \epsilon}{\simeq} \frac{\sqrt{\epsilon}}{4} \tag{42}$$

Figure 10 summarizes the results seen in this section and the previous section. We have plotted the variance of $P$ conditioned to the integrated signal and the variance of $P$ to show the reduction in variance obtained in each regime as a function of the squeezing time.

---

[4]In [23], where $\epsilon = 0$, we find the same scaling laws for $\tilde{\gamma}$ but with different numerical coefficients since it is $\mathcal{V}(\tau)$ that is minimized there.

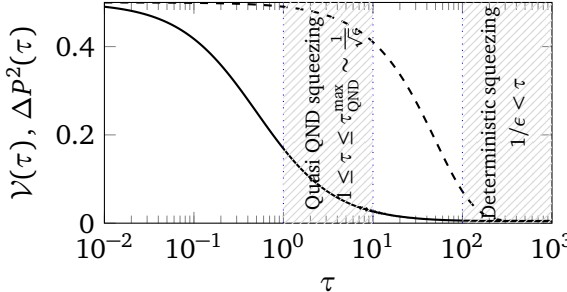

Figure 10: Time evolution of $\mathcal{V}(\tau) = \mathrm{Var}_{\sigma=\mathcal{S}}(P)$ variance of $P$ conditioned to the signal (solid line) and of $\Delta P^2$ the variance of $P$ (dashed line). The hatched areas correspond to the two possible squeezing regimes: quasi-QND, deterministic. Parameters: $\epsilon = 10^{-2}$; $\tilde{\gamma} = 0$.

## 5  Application to helium 3

The theory we have developed in the previous sections also applies, with minor modifications detailed in this section, to the generation of squeezed nuclear spin states of helium atoms in their ground state. The ground state of helium, separated by 20 eV from the first excited state, is difficult to access directly by laser. To manipulate it, using a discharge, a small fraction of the atoms $\sim 10^{-6}$ is maintained in a metastable state, the state $2^3S$, which can be coupled to light on the one hand, and which is coupled to the ground state via so-called metastability exchange collisions on the other hand [48]. In a previous work, we studied the possibility of squeezing the purely nuclear collective spin of a gas of helium-3 atoms in a cell at room temperature by quantum non-demolition measurement performed in the metastable state $2^3S$, using the Faraday effect on a transition $f = 1/2 \rightarrow f' = 1/2$ from the line $2^3S \rightarrow 2^3P$ at 1083 nm [23, 24]. More recently, a second configuration was identified, on a transition $f = 3/2 \rightarrow f' = 5/2$ of the same line, which has the advantage of working for a completely polarized atomic spin state (nuclear and metastable). The effectiveness of this configuration at the semi-classical level was demonstrated by looking at the effective coupling between classical fluctuations of the nuclear spin and the optical signal [25]. However, the question remained open regarding the complete quantum treatment of the problem, including in particular the tensorial part of the interaction between light and the metastable level $f = 3/2$ and its effect on spin squeezing.

### 5.1  Model Hamiltonian and 3-mode master equation

We consider a helium 3 gas initially polarized along $x$ by optical pumping, and interacting, for the fraction of atoms in the metastable state, with a cavity mode also polarized along $x$ and propagating along $z$. The vector coupling constant $\alpha^v$ and tensor coupling constant $\alpha^t$ determining $\Omega_V$ and $\Omega_T$ (see equations (15)), due to transitions from state $2^3S$ $f = 3/2$ to states $2^3P$, are shown in Figure 11 as a function of atomic detuning, counted from the so-called C3 transition, $f = 3/2 \rightarrow f' = 5/2$. For a frequency detuning of the order of $-2$GHz (see inset in Figure 11), $\alpha_t$ vanishes. From the linearized semi-classical equations describing the dynamics of light and atomic variable fluctuations around the fully polarized steady state (equations (78) in [25] with $M = 1$), by applying the Holstein-Primakoff approximation extended to atomic vector and tensor operators as well as to light Stokes operators, we obtain a coupled system for the evolution of the six quadratures of three bosonic modes: a mode $X_c, P_c$ for the fluctuations of the Stokes spin in the cavity, a mode $X_I, P_I$ for the fluctuations of the nuclear spin $I = 1/2$ of the ground state, and a mode $X, P$ for the fluctuations of the spin

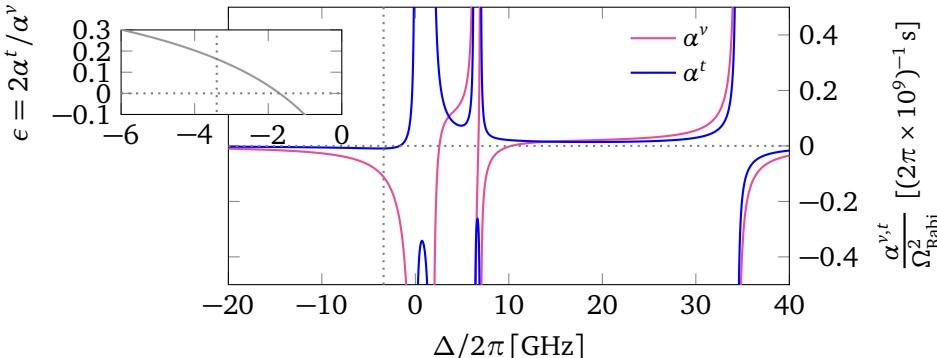

Figure 11: Vectorial and tensorial couplings in $(2\pi \times 10^9)^{-1}$ s for the polarized state $F = 3/2$, i.e. on the transition $2^3S \to 2^3P$, as a function of the frequency detuning for $^3$He. Top left, plot of $\epsilon = 2\alpha^t/\alpha^v$: the dotted vertical line marks the ideal laser frequency to minimize the perturbation induced by the tensor term, while avoiding spontaneous emission (not shown here). The frequencies are relative to the C3 transition.

373   $f = 3/2$ of the metastable state [5].

$$
\begin{pmatrix} \dot{X}_c \\ \dot{P}_c \\ \dot{X}_I \\ \dot{P}_I \\ \dot{X} \\ \dot{P} \end{pmatrix} = \begin{pmatrix} -\kappa/2 & \tilde{\delta} - \frac{3}{2}\alpha^t n & 0 & 0 & 0 & \alpha^v \frac{\sqrt{3 n_{ph} n}}{2} \\ \frac{3}{2}\alpha^t n - \tilde{\delta} & -\kappa/2 & 0 & 0 & -\alpha^t\sqrt{3 n_{\mathrm{ph}} n} & 0 \\ 0 & 0 & -\gamma_f & B_x \gamma_{\mathrm{nuc}} & \gamma_m \sqrt{\frac{n}{3N}} & 0 \\ 0 & 0 & -B_x \gamma_{\mathrm{nuc}} & -\gamma_f & 0 & \gamma_m \sqrt{\frac{n}{3N}} \\ 0 & \alpha^v \frac{\sqrt{3 n_{ph} n}}{2} & \gamma_m \sqrt{\frac{n}{3N}} & 0 & -\frac{\gamma_m}{3} & \gamma_{\frac{3}{2}} B_x - 2\alpha^t n_{\mathrm{ph}} \\ -\alpha^t\sqrt{3 n_{\mathrm{ph}} n} & 0 & 0 & \gamma_m \sqrt{\frac{n}{3N}} & 2\alpha^t n_{\mathrm{ph}} - \gamma_{\frac{3}{2}} B_x & -\frac{\gamma_m}{3} \end{pmatrix} \begin{pmatrix} X_c \\ P_c \\ X_I \\ P_I \\ X \\ P \end{pmatrix} \tag{43}
$$

374   In the matrix (43) of the equations of motion, $n$ and $N$ denote the number of atoms in the
375   metastable state and the ground state, $\gamma_m$ and $\gamma_f \equiv \gamma_m \frac{n}{N}$ are the exchange collision rates of
376   metastability for an atom in the metastable state and for an atom in the ground state, $n_{\mathrm{ph}}$ is
377   the number of photons in the linearly polarized cavity mode along $x$, $B_x$ a magnetic field along
378   $x$, $\gamma_{\mathrm{nuc}}$ and $\gamma_{\frac{3}{2}}$ the gyromagnetic factors for the nuclear spin and for the spin $f = 3/2$ of the
379   metastable state. From the semi-classical equations linearized on the quadratures (43), using
380   a standard procedure in quantum optics [23], we can write a master equation for the three
381   corresponding bosonic modes, the light mode, the nuclear mode, and the metastable mode:

$$
\frac{d\rho}{dt} = \frac{1}{i\hbar}[H, \rho] + \kappa\left(c\rho c^\dagger - \frac{1}{2}\{c^\dagger c, \rho\}\right) + C_m \rho C_m^\dagger - \frac{1}{2}\{C_m^\dagger C_m, \rho\} \tag{44}
$$

382   The Hamiltonian $H$ of interaction between light and the metastable level in (44) corresponds,
383   for $f = 3/2$, to the Hamiltonian (13) calculated in the general case in section 2, where this
384   time the ground level corresponds to the metastable level and the excited level to the level
385   $2^3P$. For $\tilde{\delta} = \frac{3}{2}\alpha^t n$ and $\gamma_{\frac{3}{2}} B_x = 2\alpha^t n_{\mathrm{ph}}$, we find equation (17):

$$
H = \hbar\Omega_V(PP_c + \epsilon X X_c) \tag{45}
$$

---

[5]Remarkably, and in accordance with the results in section 2, the other bosonic modes derived from the atomic tensor operators of the metastable states, which are coupled to each other and to the nuclear mode by metastability exchange collisions, are decoupled from the three modes of interest to us.

where we have introduced the coupling constants defined in equation (15), applied here with $f = 3/2$:

$$\Omega_V = \alpha^v \frac{\sqrt{3n_{ph}n}}{2} \qquad \Omega_T = \alpha^t \sqrt{3n_{ph}n} \qquad \epsilon = \Omega_T/\Omega_V \qquad (46)$$

As in [23,24], in addition to the jump operator $\sqrt{\kappa}c$ describing the exit of photons from the cavity, in (44) there is a jump operator for the exchange of metastability $C_m = \sqrt{2\gamma_f}a_I - \sqrt{2\gamma_m}a$ where $a_I$ and $a$ are the annihilation operators of an excitation in the nuclear and metastable modes, respectively. The final form of the three-mode master equation is obtained by introducing the eigenmodes $\alpha$ and $\beta$ of the exchange collisions, which are hybrid states between the nuclear mode $a_I$ and the metastable mode $a$ [23]:

$$\alpha = \sqrt{\frac{\gamma_m}{\gamma_m + \gamma_f}}\, a_I + \sqrt{\frac{\gamma_f}{\gamma_m + \gamma_f}}\, a \qquad \beta = \sqrt{\frac{\gamma_m}{\gamma_m + \gamma_f}}\, a - \sqrt{\frac{\gamma_f}{\gamma_m + \gamma_f}}\, a_I \qquad (47)$$

Mode $\alpha$, which is essentially nuclear, is slow, while mode $\beta$, which is essentially metastable, is fast. In this basis, the metastability exchange jump operator is reduced to mode $\beta$ alone with a rate $\gamma_\beta \equiv 2(\gamma_m + \gamma_f)$, and the master equation becomes:

$$\frac{d\rho}{dt} = \frac{1}{i\hbar}[H,\rho] + \kappa\left(c\rho c^\dagger - \frac{1}{2}\{c^\dagger c, \rho\}\right) + \gamma_\beta\left(\beta\rho\beta^\dagger - \frac{1}{2}\{\beta^\dagger\beta, \rho\}\right) \qquad (48)$$

where the Hamiltonian $H$ is now expressed in terms of hybrid modes $\alpha$ and $\beta$:

$$H = \hbar(\Omega_{V\alpha}P_\alpha + \Omega_{V\beta}P_\beta)P_c + \hbar(\Omega_{T\alpha}X_\alpha + \Omega_{T\beta}X_\beta)X_c \qquad (49)$$

with:

$$\Omega_{V\alpha} = \sqrt{\frac{\gamma_f}{\gamma_f + \gamma_m}}\Omega_V \quad \text{et} \quad \Omega_{V\beta} = \sqrt{\frac{\gamma_m}{\gamma_f + \gamma_m}}\Omega_V \qquad (50)$$

$$\Omega_{T\alpha} = \sqrt{\frac{\gamma_f}{\gamma_f + \gamma_m}}\Omega_T \quad \text{et} \quad \Omega_{T\beta} = \sqrt{\frac{\gamma_m}{\gamma_f + \gamma_m}}\Omega_T \qquad (51)$$

Compared to [23], the master equation (48) has an additional term in $H$, the term in $X_c$ of equation (49).

## 5.2 Deterministic nuclear spin squeezing of helium 3

In the same way as in section 3, we look at the possibility of deterministic squeezing of the nuclear spin from the three-mode master equation (48). After calculation, we obtain two closed systems of six equations for the second moments, one system determining the fluctuations of the nuclear spin quadrature $X_\alpha$ and another for those of the quadrature $P_\alpha$:

$$
\begin{cases}
\dfrac{d}{dt}\langle X_\alpha^2\rangle = 2\Omega_{V\alpha}\langle X_\alpha P_c\rangle \\[2mm]
\dfrac{d}{dt}\langle X_\beta^2\rangle = -\gamma_\beta(\langle X_\beta^2\rangle - \frac{1}{2}) + 2\Omega_{V\beta}\langle X_\beta P_c\rangle \\[2mm]
\dfrac{d}{dt}\langle X_\alpha P_c\rangle = -\dfrac{\kappa}{2}\langle X_\alpha P_c\rangle - \Omega_{T\beta}\langle X_\alpha X_\beta\rangle - \Omega_{T\alpha}\langle X_\alpha^2\rangle + \Omega_{V\alpha}\langle P_c^2\rangle \\[2mm]
\dfrac{d}{dt}\langle X_\beta P_c\rangle = -\dfrac{\gamma_\beta + \kappa}{2}\langle X_\beta P_c\rangle - \Omega_{T\alpha}\langle X_\alpha X_\beta\rangle - \Omega_{T\beta}\langle X_\beta^2\rangle + \Omega_{V\beta}\langle P_c^2\rangle \\[2mm]
\dfrac{d}{dt}\langle X_\alpha X_\beta\rangle = -\dfrac{\gamma_\beta}{2}\langle X_\alpha X_\beta\rangle + \Omega_{V\beta}\langle X_\alpha P_c\rangle + \Omega_{V\alpha}\langle X_\beta P_c\rangle \\[2mm]
\dfrac{d}{dt}\langle P_c^2\rangle = -\kappa(\langle P_c^2\rangle - \frac{1}{2}) - 2\Omega_{T\alpha}\langle X_\alpha P_c\rangle - 2\Omega_{T\beta}\langle X_\beta P_c\rangle
\end{cases}
\qquad (52)
$$

$$
\begin{cases}
\dfrac{d}{dt}\langle P_\alpha^2\rangle = -2\Omega_{T\alpha}\langle P_\alpha X_c\rangle \\[2mm]
\dfrac{d}{dt}\langle P_\beta^2\rangle = -\gamma_\beta(\langle P_\beta^2\rangle - \dfrac{1}{2}) - 2\Omega_{T\beta}\langle P_\beta X_c\rangle \\[2mm]
\dfrac{d}{dt}\langle P_\alpha X_c\rangle = -\dfrac{\kappa}{2}\langle P_\alpha X_c\rangle + \Omega_{V\beta}\langle P_\alpha P_\beta\rangle + \Omega_{V\alpha}\langle P_\alpha^2\rangle - \Omega_{T\alpha}\langle X_c^2\rangle \\[2mm]
\dfrac{d}{dt}\langle P_\beta X_c\rangle = -\dfrac{\gamma_\beta + \kappa}{2}\langle P_\beta X_c\rangle + \Omega_{V\alpha}\langle P_\alpha P_\beta\rangle + \Omega_{V\beta}\langle P_\beta^2\rangle - \Omega_{T\beta}\langle X_c^2\rangle \\[2mm]
\dfrac{d}{dt}\langle P_\alpha P_\beta\rangle = -\dfrac{\gamma_\beta}{2}\langle P_\alpha P_\beta\rangle - \Omega_{T\beta}\langle P_\alpha X_c\rangle - \Omega_{T\alpha}\langle P_\beta X_c\rangle \\[2mm]
\dfrac{d}{dt}\langle X_c^2\rangle = -\kappa(\langle X_c^2\rangle - \dfrac{1}{2}) + 2\Omega_{V\alpha}\langle P_\alpha X_c\rangle + 2\Omega_{V\beta}\langle P_\beta X_c\rangle
\end{cases}
\tag{53}
$$

Each of the systems (52) and (53) admits a stationary solution, whose exact expressions for the second moments are given in Appendix E. Here, we give their expressions at order $\epsilon$ and in case $\gamma_\beta, \kappa \gg \Omega_V$:

$$
\langle X_\alpha^2\rangle_{\text{deter}} = \frac{1}{2\epsilon} - \frac{2\Omega_V^2}{\gamma_\beta \kappa} + \frac{2\epsilon\Omega_V^2}{\gamma_\beta \kappa}
\qquad\qquad
\langle P_\alpha^2\rangle_{\text{deter}} = \frac{\epsilon}{2} + \frac{2\epsilon\Omega_V^2}{\gamma_\beta \kappa}
$$

$$
\langle X_\beta^2\rangle_{\text{deter}} = \frac{1}{2} + \frac{2\Omega_V^2}{\gamma_\beta(\gamma_\beta + \kappa)} - 2\Omega_V^2\,\epsilon\,\frac{4\Omega_V^2 + \gamma_\beta\kappa}{\gamma_\beta^2\kappa(\gamma_\beta + \kappa)}
\qquad
\langle P_\beta^2\rangle_{\text{deter}} = \frac{1}{2} - \epsilon\frac{2\Omega_V^2}{\gamma_\beta(\gamma_\beta + \kappa)}
\tag{54}
$$

$$
\langle X_c^2\rangle_{\text{deter}} = \frac{1}{2} + \frac{2\Omega_V^2}{\kappa(\gamma_\beta + \kappa)} - 2\Omega_V^2\,\epsilon\,\frac{4\Omega_V^2 + \gamma_\beta\kappa}{\gamma_\beta\kappa^2(\gamma_\beta + \kappa)}
\qquad
\langle P_c^2\rangle_{\text{deter}} = \frac{1}{2} - \epsilon\frac{2\Omega_V^2}{\kappa(\gamma_\beta + \kappa)}
$$

which clearly demonstrates the existence of a deterministic squeezing regime with a reduction in quantum noise on the hybrid quadrature $P_\alpha$ of the order of $\epsilon/2$, as in the two-mode case (section 3). To obtain nuclear spin squeezing, it suffices to return to the basis $a, b$:

$$
P_I = \sqrt{\frac{\gamma_m}{\gamma_m + \gamma_f}}\,P_\alpha - \sqrt{\frac{\gamma_f}{\gamma_m + \gamma_f}}\,P_\beta
\tag{55}
$$

hence the noise reduction on $P_I$:

$$
\Delta P_I^2 = \frac{\gamma_m}{\gamma_m + \gamma_f}\Delta P_\alpha^2 + \frac{\gamma_f}{\gamma_m + \gamma_f}\Delta P_\beta^2 \simeq \Delta P_\alpha^2
\qquad\text{as}\qquad
\frac{\gamma_f}{\gamma_m} = \frac{n}{N} \simeq 10^{-6}
\tag{56}
$$

To find the characteristic time for deterministic squeezing, we eliminate adiabatically the fast components in systems (52) and (53). In each of these two systems, the last five equations have a damping term, unlike the first. We can therefore seek quasi-stationary solutions for the damped variables expressed as a function of the slow variable $\langle X_\alpha^2\rangle$ (resp. $\langle P_\alpha^2\rangle$) and the problem parameters. In particular, the solution obtained for $\langle X_\alpha P_c\rangle$ (resp. $\langle P_\alpha X_c\rangle$) can be fed back in order to obtain a first-order differential equation for $\langle X_\alpha^2\rangle$ (resp. $\langle P_\alpha^2\rangle$). After calculation, we obtain [6], still at order $\epsilon$:

$$
\langle X_\alpha^2\rangle(\tau) \simeq \langle X_\alpha^2\rangle_{\text{deter}} - \left(\langle X_\alpha^2\rangle_{\text{deter}} - \frac{1}{2}\right)e^{-2\epsilon\tau}
\quad;\quad
\langle P_\alpha^2\rangle(\tau) \simeq \langle P_\alpha^2\rangle_{\text{deter}} + \left(\frac{1}{2} - \langle P_\alpha^2\rangle_{\text{deter}}\right)e^{-2\epsilon\tau}
\tag{57}
$$

$$
\text{with}\quad \tau \equiv \Gamma_\alpha t \qquad \Gamma_\alpha \equiv \frac{2\Omega_{V\alpha}^2}{\kappa} \simeq \frac{\gamma_f}{\gamma_m}\Gamma
\tag{58}
$$

In Figure 12, we have plotted the time evolution of the quantum fluctuations of quadratures $P_\alpha$ and $X_\alpha$ in case $\epsilon < 1$ for which $P_\alpha$ is squeezed.

---

[6]The equations obtained for three modes have the same form as in section 3 equation (23) for two modes.

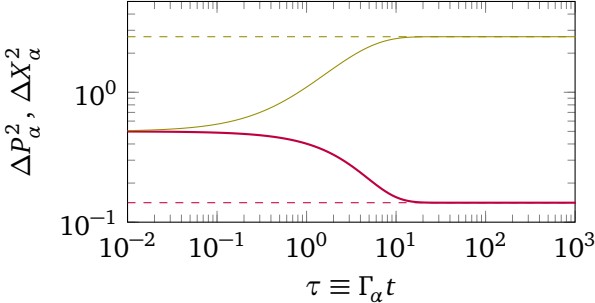

Figure 12: Deterministic nuclear squeezing in ${}^3\mathrm{He}$ from the three-mode model: time evolution of quantum fluctuations of hybrid quadratures $P_\alpha$ (decreasing curve in red) and $X_\alpha$ (increasing curve in green). The expressions for the asymptotic limits are given in Appendix E. Parameters (see section 5.4): $\epsilon = 1.6 \times 10^{-1}$; $\gamma_\beta/\Omega_V = 0.48$; $\kappa/\Omega_V = 39$.

### 5.3 Nuclear spin squeezing of ${}^3\mathrm{He}$ by continuous homodyne detection

#### 5.3.1 Master equation for the nuclear mode

Analogous to section 4.1, but this time with two strongly damped modes, the cavity mode $c$ and the metastable hybrid mode $\beta$, we adiabatically eliminate these two modes using the Monte Carlo wave function formalism [44]. From the master equation (48), we deduce the effective Hamiltonian:

$$H_{\text{eff}} = \hbar(\Omega_{V\alpha}P_\alpha + \Omega_{V\beta}P_\beta)P_c + \hbar(\Omega_{T\alpha}X_\alpha + \Omega_{T\beta}X_\beta)X_c - \frac{i\hbar}{2}\kappa c^\dagger c - \frac{i\hbar}{2}\gamma_\beta \beta^\dagger \beta \tag{59}$$

In the weak coupling limit $\Omega_V \to 0$, we can truncate the Monte Carlo state vector in the Fock basis $|\psi\rangle = \sum_{n_\beta,n_c} \left|\psi_\alpha^{n_\beta n_c}\right\rangle_{\text{nuc}} |n_\beta\rangle_{\text{meta}} |n_c\rangle_{\text{cav}}$, at first order in $\Omega_V$ (i.e. by a single action of the effective Hamiltonian), as follows:

$$|\psi\rangle = \left|\psi_\alpha^{00}\right\rangle |0\rangle |0\rangle + \left|\psi_\alpha^{01}\right\rangle |0\rangle |1\rangle + \left|\psi_\alpha^{11}\right\rangle |1\rangle |1\rangle \tag{60}$$

Under the effect of the effective Hamiltonian, the two fast components $\left|\psi_\alpha^{01}\right\rangle$ and $\left|\psi_\alpha^{11}\right\rangle$ exponentially join an adiabatic following regime of the slow component $\left|\psi_\alpha^{00}\right\rangle$, hence their adiabatic elimination:

$$\begin{aligned}
\left|\psi_\alpha^{01}\right\rangle &\simeq \frac{\sqrt{2}}{\kappa}\left(\Omega_{V_\alpha}P_\alpha - i\Omega_{T_\alpha}X_\alpha\right)\left|\psi_\alpha^{00}\right\rangle \\
\left|\psi_\alpha^{11}\right\rangle &\simeq -i\frac{\Omega_{T_\beta} - \Omega_{V_\beta}}{\kappa + \gamma_\beta}\left|\psi_\alpha^{00}\right\rangle
\end{aligned} \tag{61}$$

Substituting these expressions into the Schrödinger equation verified by $\left|\psi_\alpha^{00}\right\rangle$:

$$i\hbar\frac{d\left|\psi_\alpha^{00}\right\rangle}{dt} = -\frac{i\hbar}{2}\left(\Gamma_\alpha C^\dagger C + \Gamma_0\right)\left|\psi_\alpha^{00}\right\rangle \equiv H_{\text{eff}}^{00}\left|\psi_\alpha^{00}\right\rangle \tag{62}$$

where $C = P_\alpha - i\epsilon X_\alpha$ has the same form as the non-Hermitian jump operator (31) (but for mode $\alpha$), $\Gamma_\alpha$ has been defined in (58), and $\Gamma_0 = \frac{(\Omega_{T_\beta} - \Omega_{V_\beta})^2}{\kappa + \gamma_\beta}$. We can therefore write a master equation:

$$\frac{d\rho^0}{dt} = \Gamma_\alpha\left(C\rho^0 C^\dagger - \frac{1}{2}\left\{C^\dagger C, \rho^0\right\}\right) + \Gamma_0\left(C_d\rho^0 C_d^\dagger - \frac{1}{2}\left\{C_d^\dagger C_d, \rho^0\right\}\right) \tag{63}$$

in terms of two quantum jumps $C$ and $C_d = \mathbb{1}$. Compared to the structure of master equation (30), a jump term proportional to the identity is added, which will not play a role in homodyne detection measurement.

### 5.3.2  Nuclear spin squeezing by continuous homodyne measurement

As in section 4.2, we can associate with each jump operator of the master equation (63) a stochastic process with real values, Gaussian, with zero mean, variance dt and no memory. The contribution of jump $\sqrt{\Gamma_0}C_d$ in the stochastic reformulation of the master equation (see equation (55) in [23]) is zero. Thus, the theoretical results (41) and (42) concerning squeezing by continuous homodyne measurement seen in section 4.2.2 apply directly, with and without decoherence, and allow the squeezing of the hybrid quadrature $P_\alpha$ to be quantified. As in (56):

$$\text{Var}_{\sigma=\mathcal{S}}(P_I) = \frac{\gamma_m}{\gamma_m + \gamma_f}\text{Var}_{\sigma=\mathcal{S}}(P_\alpha) + \frac{\gamma_f}{\gamma_m + \gamma_f}\langle P_\beta^2 \rangle \simeq \text{Var}_{\sigma=\mathcal{S}}(P_\alpha) = \mathcal{V}(\tau) \tag{64}$$

### 5.4  Numerical estimates

Numerical values of the parameters for a non-demolition Faraday quantum spin squeezing experiment in a cell of approximately $400\,\text{mm}^3$ filled with $^3$He at room temperature were given in reference [24]. For a pressure of 0.88 Torr, there would be approximately $N = 10^{16}$ atoms in the ground state and $n = 5 \times 10^{10}$ atoms in the metastable state in the steady state in the presence of a discharge [48]. As in [24], we consider the system in a cavity pumped with linearly polarized light in the $x$ direction, with a loss rate of the number of photons in the cavity $\kappa = 2\pi \times 10^8$Hz and 5mW of $x$-polarized light leaving the cavity.

Unlike [23, 24], where the light was tuned to a transition $F = 1/2 \to F' = 1/2$ and the atomic system was only partially polarized, we consider here a more favorable configuration [25], which uses the hyperfine level $F = 3/2$ of the metastable state and a completely polarized atomic system, resulting in the metastability exchange collision rates $\gamma_m = 3.92 \times 10^6\,\text{s}^{-1}$ and $\gamma_f = 19.6\,\text{s}^{-1}$. For an atom-light detuning of $-3.4$GHz, corresponding to the dotted line on the inset of Figure 11, for which $\epsilon = 0.16$, we would then have: $n_{\text{ph}} = 4.33 \times 10^7$, $\Omega_{\text{Rabi}} = 2\pi \times 4.26$ kHz, $\Omega_V = 2\pi \times 2.59$ MHz, and $\Gamma_\alpha = 4.22\,\text{s}^{-1}$. The decoherence in the ground state, dominated by the de-excitation of the metastable states on the walls of the cell [24], would give a parameter $\tilde{\gamma}_\alpha = \gamma/\Gamma_\alpha = 3.08 \times 10^{-2}$ here.

For deterministic spin squeezing, according to Figure 12 obtained without decoherence, for $\tau = 10$, we could have $\Delta P^2/\Delta P_0^2 \simeq 0.2$ in a time of 2.37s. In the presence of decoherence, the correction (see equation (23)) is not negligible here since $\tilde{\gamma}_\alpha/2\epsilon \simeq 9.6 \times 10^{-2}$ and we would then obtain $\Delta P^2/\Delta P_0^2 \simeq 0.24$.

For squeezing by continuous measurement, according to equations (42), we could achieve a conditional variance $\mathcal{V}_{\text{QND}}/\mathcal{V}(\text{t}=0) \simeq 0.2$ in a time $t_{\text{QND}} = \tau_{\text{QND}}/\Gamma_\alpha = 0.59$s. With the corrections (41) due to decoherence, $\tilde{\gamma}_\alpha/6 \simeq 5 \times 10^{-3}$, we would achieve $\mathcal{V}_{\text{QND}}/\mathcal{V}(\text{t}=0) \simeq 0.22$. An interesting avenue for reducing atomic decoherence, which would also allow the nuclear spin size to be varied for a constant pressure in the cell, would be to use a mixture of helium 3 and helium 4 instead of pure helium 3 gas [48].

## 6  Conclusion

In this article, we propose using both vector and tensor light shifts on spin-$f > 1/2$ atoms to squeeze the transverse fluctuations of the collective atomic spin starting from a polarized state. For an atomic spin and a Stokes spin of light polarized in the same direction, we derive a simple Hamiltonian in terms of quadratures, which allows us to obtain analytical results as a function of the ratio $\epsilon$ between the vector and tensor coupling of the atom-field interaction. For $0 < \epsilon \ll 1$, we identify two distinct regimes of spin squeezing. A regime of squeezing by quasi-QND measurement using the Faraday effect for times up to $(\sqrt{\epsilon}\Gamma)^{-1}$, where $\Gamma$ is the usual QND measurement rate by Faraday effect, where the conditional variance of the

transverse fluctuations of the atomic spin is reduced by a factor of $\sqrt{\epsilon}/2$, and a second regime of deterministic spin squeezing for times of the order of $(\epsilon\Gamma)^{-1}$, where the variance of the fluctuations is reduced by a factor of $\epsilon$. The advantage of the deterministic squeezing regime is to prepare an unconditional state and to squeeze better, by a factor $\sqrt{\epsilon}$. The advantage of the quasi-QND regime, holding for times $\Gamma^{-1} \ll t < (\sqrt{\epsilon}\Gamma)^{-1}$, is that the squeezing is faster, which could be needed in some cases to maintain the decoherence rate lower than the squeezing rate. If our analysis is general and can be applied to different atomic species, we apply it here to the atoms of $^{87}$Sr, $^{173}$Yb, and $^{3}$He, all of which have a purely nuclear spin in the ground state with very long associated coherence times. In a cavity such as that of [1], in the deterministic squeezing regime, it would be possible to obtain for strontium 87 a reduction in variance of 0.35 in a time of the order of a second, and for ytterbium 173 a reduction in variance of 0.03 in a few tens of milliseconds. In the case of helium 3, the interaction with light occurs in the metastable state $2^3S_1$ of spin $f = 3/2$, and spin squeezing is brought back to the ground state $1^1S_0$ of spin 1/2 thanks to metastability exchange collisions. For this atom, in continuous measurement squeezing, it would then be possible to achieve a reduction in variance of 0.22 in 0.59s. Spin squeezing in these atoms thus offers significant prospects for metrology, whether in magnetometry or atomic clocks.

# Acknowledgements

We would like to thank Romain Long, Jakob Reichel, Pierre-Jean Nacher and Yvan Castin for fruitful discussions.

# A Atom-photon interaction and light shifts in the ground state

Here we are interested in the lightshift in the ground state for an atom placed in an electromagnetic field at frequency $\omega$. For a monochromatic light field, the atom-electric dipole field interaction is written as [32, 33]:

$$V = -\boldsymbol{d} \cdot \boldsymbol{E} \tag{A.1}$$

where $\boldsymbol{d}$ denotes the atomic dipole operator and $\boldsymbol{E}$ the electric field operator. Since we are interested in the complete quantum treatment of the atom-photon system, we also quantize the field in a cavity of volume $\mathcal{V}$. Taking polarization into account, we arbitrarily note a horizontal linear polarization with direction vector $\mathbf{e}_{\mathrm{H}}$ and a vertical polarization with direction vector $\mathbf{e}_{\mathrm{V}}$ orthogonal to $\mathbf{e}_{\mathrm{H}}$, with the beam propagating in direction $\mathbf{e}_k \equiv \mathbf{e}_{\mathrm{H}} \wedge \mathbf{e}_{\mathrm{V}}$. The field operator is then written, for component $(+)$:

$$\mathbf{E}^{(+)} = \sqrt{\frac{\hbar\omega}{2\epsilon_0 \mathcal{V}}} (a_{\mathrm{H}}\mathbf{e}_{\mathrm{H}} + a_{\mathrm{V}}\mathbf{e}_{\mathrm{V}}) \tag{A.2}$$

where $a_{\mathrm{H}}$ and $a_{\mathrm{V}}$ are the photon annihilation operators for the cavity mode of frequency $\omega_c$. To these operators, we can associate the Stokes vector $\mathbf{S}$, a collective operator that is equivalent to a spin $n_{\mathrm{ph}}/2$, where $n_{\mathrm{ph}}$ is the number of photons, to represent the degrees of freedom of the field polarization, and whose components are written as:

$$S_1 = \frac{1}{2}(a_{\mathrm{H}}^\dagger a_{\mathrm{H}} - a_{\mathrm{V}}^\dagger a_{\mathrm{V}}) \; ; \; S_2 = \frac{1}{2}(a_{\mathrm{H}}^\dagger a_{\mathrm{V}} + a_{\mathrm{V}}^\dagger a_{\mathrm{H}}) \; ; \; S_3 = \frac{1}{2i}(a_{\mathrm{H}}^\dagger a_{\mathrm{V}} - a_{\mathrm{V}}^\dagger a_{\mathrm{H}}) \tag{A.3}$$

As with angular momentum, these operators satisfy the commutation relations $[S_i, S_j] = i\epsilon_{ijk}S_k$. We also define $S_0 = \frac{1}{2}(a_{\mathrm{H}}^\dagger a_{\mathrm{H}} + a_{\mathrm{V}}^\dagger a_{\mathrm{V}})$; $2S_0$ is therefore the photon number operator. In the rotating wave approximation, and using a perturbative approach for a highly detuned laser field, the effective Hamiltonian of the particle in the ground state can be written as follows [32, 33]:

$$h_f = \sum_e \boldsymbol{E}^{(-)} \cdot \frac{\boldsymbol{\alpha}_{g,e}}{\hbar\Delta_{g,e}} \cdot \boldsymbol{E}^{(+)} \tag{A.4}$$

where $\boldsymbol{\alpha}_{g,e}$ denotes the atomic polarizability tensor, $g$ and $e$ correspond respectively to the ground state of the spin $f$ atom and to a level of the excited state with spin $f'$, and $\Delta_{g,e} = \omega - \omega_{eg}$ is the detuning between the frequency of the light field and an atomic resonance frequency $\omega_{eg} \equiv \omega_e - \omega_g$. The tensor $\boldsymbol{\alpha}_{g,e}$ is written as:

$$\boldsymbol{\alpha}_{g,e} = \boldsymbol{d}_{ge}\boldsymbol{d}_{eg} \tag{A.5}$$

where $\boldsymbol{d}_{ge} = \boldsymbol{d}_{eg}^\dagger = P_g \boldsymbol{d} P_e$ is the electric dipole operator that allows the atomic transition from state $(e)$ to state $(g)$. Physically, this interaction Hamiltonian (A.4) means that, starting from a fundamental state, the atom is brought to an excited (virtual) state by absorbing a photon from the field, which is well described by the coupling operation between the scale operator $\boldsymbol{d}_{eg}$ and the annihilation of a photon via $\boldsymbol{E}^{(+)}$. The temporarily excited atom then returns to its ground state (potentially another $(g)$) by emitting a photon scattered in the light field, i.e. via the coupling between $\boldsymbol{d}_{ge}$ and $\boldsymbol{E}^{(-)}$.

Since the polarizability tensor (A.5) is a rank 2 spherical tensor (as the dyadic sum of two vectors $\boldsymbol{d}$ and $\boldsymbol{d}^\dagger$, which are rank 1 spherical tensors), it can be decomposed irreducibly into spherical components, as can the Hamiltonian (A.4). The total Hamiltonian is the sum over all permitted transitions $g \rightarrow e$:

$$h_f = \sum_e \frac{\hbar\omega}{2\epsilon_0 \mathcal{V}} \frac{\alpha_0}{\hbar\Delta_{g,e}} \left\{ \alpha_{g,e}^{(v)} f_k S_3 + \alpha_{g,e}^{(t)} \left[ \left( \frac{f(f+1)}{3} - f_k^2 \right) S_0 + (f_{\mathrm{H}}^2 - f_{\mathrm{V}}^2)S_1 + (f_{\mathrm{H}}f_{\mathrm{V}} + f_{\mathrm{V}}f_{\mathrm{H}})S_2 \right] \right\} \tag{A.6}$$

where the $f_k, f_H, f_V$ are the Cartesian components of the spin $f$ of the atom, $\alpha_0$ is a character- istic constant proportional to $\gamma_{sp}$ the spontaneous emission rate of the wavelength transition $\lambda$:

$$\alpha_0 = \frac{3\epsilon_0 \hbar \gamma_{sp} \lambda^3}{8\pi^2} \tag{A.7}$$

and $\alpha_{g,e}^{(v)}, \alpha_{g,e}^{(t)}$ are constants depending solely on the quantum numbers of the transition under consideration:

$$\alpha_{g,e}^{(v)} = (2j'+1) \left| \begin{pmatrix} 1 & j & j' \\ i & f' & f \end{pmatrix} \right|^2 \left( -\frac{2f-1}{f}\delta_{f-1}^{f'} - \frac{2f+1}{f(f+1)}\delta_f^{f'} + \frac{2f+3}{f+1}\delta_{f+1}^{f'} \right) \tag{A.8}$$

$$\alpha_{g,e}^{(t)} = -(2j'+1) \left| \begin{pmatrix} 1 & j & j' \\ i & f' & f \end{pmatrix} \right|^2 \left( \frac{1}{f}\delta_{f-1}^{f'} - \frac{2f+1}{f(f+1)}\delta_f^{f'} + \frac{1}{f+1}\delta_{f+1}^{f'} \right) \tag{A.9}$$

The matrices in parentheses correspond to Wigner symbols $6j$, and $j, j'$ are the electron spins of the ground state and the excited state. In part 2, we replaced H with $x$, V with $y$, and $k$ with $z$. We also replaced $\alpha_0$ with its expression (A.7) to reveal the effective scattering cross section $\sigma_c = \frac{3\lambda^2}{2\pi}$. Finally, $S_1, S_2, S_3$ becomes $S_x, S_y, S_z$. In the end, we obtain the expression (1), with

$$\alpha^v \equiv \frac{c\sigma_c}{4\mathcal{V}}\gamma_{sp} \sum_{e=\{j',f'\}} \frac{\alpha_{g,e}^{(v)}}{\Delta_{g,e}} \quad ; \quad \alpha^t \equiv \frac{c\sigma_c}{4\mathcal{V}}\gamma_{sp} \sum_{e=\{j',f'\}} \frac{\alpha_{g,e}^{(t)}}{\Delta_{g,e}} \tag{A.10}$$

The plots in Figures 2 and 11 represent the vector and tensor couplings divided by the square of the Rabi pulsation

$$\Omega_{\text{Rabi}}^2 = \frac{c\sigma_c}{4\mathcal{V}}\gamma_{sp} \tag{A.11}$$

# B 2-mode equations of motion close to the polarized state

In this appendix, we start from the full Hamiltonian (12) from which we write the equations of motion (quantum Langevin) from Heisenberg's point of view. The equation of motion verified by $\tilde{a}_x$ is written, in the presence of atoms:

$$\dot{\tilde{a}}_x = -i\left(\delta_c + \alpha^t \frac{nf}{3}(2f-1) - \sum_{k=1}^{2f} k(2f-k)a_k^\dagger a_k\right)\tilde{a}_x - \frac{\kappa}{2}\tilde{a}_x + \beta + d\tilde{a}_x^{\text{stoch}}/dt \tag{B.1}$$

where we have introduced $\delta_c = \omega_c - \omega$ the detuning between the laser and the cavity, $\kappa$ the damping of the light mode, and the Langevin forces (noted $d\mathcal{O}^{\text{stoch}}/dt$ for an observable $\mathcal{O}$). In the polarized state $|n : \phi_0\rangle \otimes |\alpha_x\rangle$, which is a stationary state:

$$\langle \tilde{a}_x \rangle_{\text{st}} = \frac{\beta}{\kappa/2 + i\tilde{\delta}} \tag{B.2}$$

with $\tilde{\delta} = \delta_c + \alpha^t n\frac{f}{3}(2f-1)$ the detuning in the presence of atoms in the cavity. As for the transverse operators of light and atoms, which have a zero average in the stationary polarized

state, the equations of motion are written, again based on the full Hamiltonian (12):

$$
\begin{aligned}
\dot{S}_y &= \frac{1}{2}(\beta \tilde{a}_y^\dagger + \beta^* \tilde{a}_y) - \kappa S_y + \alpha^v \sqrt{nf} P_1 S_x - \alpha^t nf(f - 1/2) S_z + dS_y^{\text{stoch}}/dt \\
\dot{S}_z &= \frac{1}{2i}(-\beta \tilde{a}_y^\dagger + \beta^* \tilde{a}_y) - \kappa S_z - \alpha^t \sqrt{nf}(2f - 1) X_1 S_x + \alpha^t nf(f - 1/2) S_y + dS_z^{\text{stoch}}/dt \\
\dot{X}_1 &= \alpha^v \sqrt{nf} S_z + \left(\gamma_f B_0 - (2f - 1)\alpha^t a_x^\dagger a_x\right) P_1 \\
\dot{P}_1 &= -\alpha^t \sqrt{nf}(2f - 1) S_y - \left(\gamma_f B_0 - (2f - 1)\alpha^t a_x^\dagger a_x\right) X_1 \\
\dot{X}_k &= \left(\gamma_f k B_0 - k(2f - k)\alpha^t a_x^\dagger a_x\right) P_k \quad , \quad k \neq 1 \\
\dot{P}_k &= -\left(\gamma_f k B_0 - k(2f - k)\alpha^t a_x^\dagger a_x\right) X_k \quad , \quad k \neq 1
\end{aligned}
\tag{B.3}
$$

By linearizing the equations of motion, using (B.2), and switching to the atomic and light quadratures introduced in (5), (6), (7), and (8), knowing that $\langle a_x^\dagger a_x \rangle_{\text{st}} = n_{\text{ph}}$:

$$
\begin{aligned}
\dot{X}_c &= \left(\tilde{\delta} - \alpha^t nf(f - 1/2)\right) P_c - \frac{\kappa}{2} X_c + \alpha^v \sqrt{\frac{n n_{\text{ph}} f}{2}} P_1 + dX_c^{\text{stoch}}/dt \\
\dot{P}_c &= -\left(\tilde{\delta} - \alpha^t nf(f - 1/2)\right) X_c - \frac{\kappa}{2} P_c - \alpha^t \sqrt{\frac{n n_{\text{ph}} f}{2}}(2f - 1) X_1 + dP_c^{\text{stoch}}/dt \\
\dot{X}_1 &= \alpha^v \sqrt{\frac{n n_{\text{ph}} f}{2}} P_c + \left(\gamma_f B_0 - (2f - 1)\alpha^t n_{\text{ph}}\right) P_1 \\
\dot{P}_1 &= -\alpha^t \sqrt{\frac{n n_{\text{ph}} f}{2}}(2f - 1) X_c - \left(\gamma_f B_0 - (2f - 1)\alpha^t n_{\text{ph}}\right) X_1 \\
\dot{X}_k &= \left(\gamma_f k B_0 - k(2f - k)\alpha^t n_{\text{ph}}\right) P_k \quad , \quad k \neq 1 \\
\dot{P}_k &= -\left(\gamma_f k B_0 - k(2f - k)\alpha^t n_{\text{ph}}\right) X_k \quad , \quad k \neq 1
\end{aligned}
\tag{B.4}
$$

$\tilde{\delta}$ has been defined in (14). From the equations of motion (B.4), we can write the Hamiltonian (13) describing the transverse fluctuations of the atomic and light modes.

# C   Interpretation of the deterministic squeezing mechanism using Langevin equations

In this appendix, instead of using the master equation formalism like in section 3, we rather use the Langevin approach, i.e. we start from equations (B.4). We also assume that $\gamma = 0$ for the sake of simplicity. By tuning $\tilde{\delta}$ and $B_0$ in equations (B.4) so that the harmonic oscillator motion of each mode is cancelled:

$$
\begin{aligned}
\dot{X}_c &= -\frac{\kappa}{2} X_c + \Omega_V P + dX_c^{\text{stoch}}/dt \\
\dot{P}_c &= -\frac{\kappa}{2} P_c - \epsilon \Omega_V X + dP_c^{\text{stoch}}/dt \\
\dot{X} &= \Omega_V P_c \\
\dot{P} &= -\epsilon \Omega_V X_c
\end{aligned}
\tag{C.1}
$$

We have omitted the index for the first atomic mode as other atomic modes are decoupled. $dX_c^{\text{stoch}}$ is a Langevin noise and for every time $t$ and $t'$, we have [23]:

$$
\langle dX_c^{\text{stoch}}(t) dX_c^{\text{stoch}}(t') \rangle = \langle dP_c^{\text{stoch}}(t) dP_c^{\text{stoch}}(t') \rangle = \frac{\kappa}{2} dt\ \delta(t - t')
\tag{C.2}
$$

The stochastic equations (C.1) allow us to derive the dynamics of the atomic mode, looking at the mean and the variance of $X$ and $P$. First, we eliminate adiabatically the light mode:

$$
\begin{aligned}
X_c &= \frac{2\Omega_V}{\kappa}P + \frac{2}{\kappa}dX_c^{\text{stoch}}/dt \\
P_c &= -\frac{2\epsilon\Omega_V}{\kappa}X + \frac{2}{\kappa}dP_c^{\text{stoch}}/dt
\end{aligned}
\tag{C.3}
$$

Reporting the expressions above into the equations of motion for the atomic mode:

$$
\begin{aligned}
\dot{X} &= -\frac{2\epsilon\Omega_V^2}{\kappa}X + \frac{2\Omega_V}{\kappa}dP_c^{\text{stoch}}/dt \\
\dot{P} &= -\frac{2\epsilon\Omega_V^2}{\kappa}X - \frac{2\epsilon\Omega_V}{\kappa}dX_c^{\text{stoch}}/dt
\end{aligned}
\tag{C.4}
$$

The stochastic equations for $X$ and $P$ directly exhibits the deterministic behaviour, decaying at a rate $\epsilon\Gamma$ for the mean $\langle X\rangle$ and $\langle P\rangle$:

$$
\begin{aligned}
\langle\dot{X}\rangle &= -\epsilon\Gamma\langle X\rangle \\
\langle\dot{P}\rangle &= -\epsilon\Gamma\langle P\rangle
\end{aligned}
\tag{C.5}
$$

As the initial conditions are such that $\langle X\rangle_{(t=0)} = \langle P\rangle_{(t=0)} = 0$, $\langle X\rangle(t) = \langle P\rangle(t) = 0$. For the variance, we thus need to look only at $\langle X^2\rangle$ and $\langle P^2\rangle$:

$$
\begin{aligned}
\frac{d}{dt}\langle P^2\rangle &= \frac{\langle(P+dP)(P+dP) - P^2\rangle}{dt} \\
&= \frac{\langle 2PdP + dP^2\rangle}{dt} \\
&= -2\epsilon\Gamma\Big(\langle P^2\rangle - \frac{\epsilon}{2}\Big)
\end{aligned}
\tag{C.6}
$$

where we have used (C.2) for $\langle dP^2\rangle$. This demonstrates the existence of a squeezing deterministic regime, provided that $\epsilon > 0$. Similarly for $\langle X^2\rangle$:

$$
\frac{d}{dt}\langle X^2\rangle = -2\epsilon\Gamma\Big(\langle X^2\rangle - \frac{1}{2\epsilon}\Big)
\tag{C.7}
$$

This also shows that if $\epsilon > 1$ then $X$ is squeezed. The deterministic spin squeezing thus derives from the coupled dynamics of the damped (cavity) and the undamped (atomic) mode. Due to the presence of the two terms $XX_c$ and $PP_c$ in the Hamiltonian, for $\epsilon > 0$ both quadratures of the atomic mode inherit the damping of the cavity mode. Due to the different weights of the two terms $XX_c$ and $PP_c$ in the hamiltonian, for $\epsilon \neq 1$, the quantum noise in the two atomic quadratures is unevenly distributed in the steady state.

# D  Calculation of the mean and variance of $P$ conditioned to the signal in the presence of decoherence

To take decoherence into account, we add a Lindblad term with a jump operator $\sqrt{\gamma}\,a$ to the master equation (30):

$$
\frac{d\rho^0}{dt} = \Gamma\Big(C\rho^0 C^\dagger - \frac{1}{2}\{C^\dagger C, \rho^0\}\Big) + \gamma\Big(a\rho^0 a^\dagger - \frac{1}{2}\{a^\dagger a, \rho^0\}\Big)
\tag{D.1}
$$

From equation (D.1), which describes the slow evolution of the atomic mode, we can write a continuous-time stochastic equation suitable for describing the evolution conditioned to a continuous measurement of a quadrature of the field leaving the cavity by homodyne detection [23, 43–45]:

$$
\begin{aligned}
d\,|\phi\rangle = & -\Gamma\frac{dt}{2}\left(C^{\dagger}C - 2\bar{p}C + \bar{p}^2\right)|\phi\rangle + \sqrt{\Gamma}d\zeta_s\,(C - \bar{p})\,|\phi\rangle \\
& -\gamma\frac{dt}{2}\left(a^{\dagger}a + i\sqrt{2}\bar{p}\,a + \frac{1}{2}\bar{p}^2\right)|\phi\rangle + \sqrt{\gamma}d\zeta_a\left(ia + \frac{\sqrt{2}}{2}\bar{p}\right)|\phi\rangle
\end{aligned}
\tag{D.2}
$$

with

$$
C^{\dagger} = P + i\epsilon X \quad ; \quad C = P - i\epsilon X \quad ; \quad \bar{p} = \langle\phi(t)|\,P\,|\phi(t)\rangle
\tag{D.3}
$$

Following [23], for the decoherence jump operator, we have chosen $\sqrt{\gamma}\,ia$ rather than $\sqrt{\gamma}\,a$, which allows us to remain in $\mathbb{R}$ for the calculations that follow. To the (non-Hermitian) jump operator $C$, we have associated a continuous-time stochastic process $d\zeta_s(t)$ with real values, Gaussian, with mean zero, variance $dt$, and no memory. Similarly, $d\zeta_a(t)$ is a continuous stochastic process with real values, Gaussian, with mean zero, variance $dt$, and no memory associated with the atomic decoherence jump operator. This equation can be solved exactly by a Gaussian ansatz for the wave function in momentum space, real and normalized to unity:

$$
\phi(p, t) = e^{-S} \quad \text{with} \quad S = u(t)\left(p - \bar{p}(t)\right)^2 - W
\tag{D.4}
$$

where $W$ represents the normalization factor. With this ansatz:

$$
\frac{d\phi}{\phi} = 2ud\bar{p}\,(p - \bar{p}) + \left(-du + 2u^2d\bar{p}^2\right)(p - \bar{p})^2 - ud\bar{p}^2 + dW
\tag{D.5}
$$

We can also rewrite equation (D.2) in $p$-representation:

$$
\begin{aligned}
\frac{d\phi}{\phi} = & -\Gamma\frac{dt}{2}\left((1 - 4u^2\epsilon^2)(p - \bar{p})^2 + 4\bar{p}u\epsilon\,(p - \bar{p}) + 2u\epsilon^2 - \epsilon\right) + \sqrt{\Gamma}d\zeta_s\,(1 - 2u\epsilon)(p - \bar{p}) \\
& -\gamma\frac{dt}{2}\left((\frac{1}{2} - 2u^2)(p - \bar{p})^2 + 2\bar{p}u\,(p - \bar{p}) + u - \frac{1}{2}\right) + \sqrt{\frac{\gamma}{2}}d\zeta_a\,(2u - 1)(p - \bar{p})
\end{aligned}
\tag{D.6}
$$

Identifying the terms in $(p - \bar{p})$, $(p - \bar{p})^2$, after calculation, we obtain the differential equations verified by $u(t)$ and $\bar{p}(t)$:

$$
\begin{aligned}
du &= \left(\Gamma(1 - 2u\epsilon) + \gamma(\frac{1}{2} - u)\right)dt \\
d\bar{p} &= -\left(\epsilon\Gamma + \frac{\gamma}{2}\right)\bar{p}\,dt + d\zeta_s\sqrt{\Gamma}\left(\frac{1}{2u} - \epsilon\right) + d\zeta_a\sqrt{\frac{\gamma}{2}}\left(1 - \frac{1}{2u}\right)
\end{aligned}
\tag{D.7}
$$

with initial conditions $u(0) = \frac{1}{2}$ and $\bar{p} = 0$, which can be solved analytically:

$$
u(\tau) = \frac{1}{2\epsilon + \tilde{\gamma}}\left(1 + \frac{\tilde{\gamma}}{2} - (1 - \epsilon)\,e^{-(2\epsilon + \tilde{\gamma})\tau}\right)
\tag{D.8}
$$

$$
\bar{p}(\tau) = e^{-(\epsilon + \frac{\tilde{\gamma}}{2})\tau}\int_0^{\tau} e^{(\epsilon + \frac{\tilde{\gamma}}{2})\tau'}w(\tau', d\zeta_s(\tau'), d\zeta_a(\tau'))
\tag{D.9}
$$

where we have set $\tau \equiv \Gamma t$, $\tilde{\gamma} \equiv \frac{\gamma}{\Gamma}$, and $w(\tau, d\zeta_s, d\zeta_a) \equiv d\zeta_s(\tau)\left(\frac{1}{2u(\tau)} - \epsilon\right) + d\zeta_a(\tau)\sqrt{\frac{\tilde{\gamma}}{2}}\left(1 - \frac{1}{2u(\tau)}\right)$.
Now let us introduce the integrated homodyne detection signal in its stochastic form:

$$
\sigma(t) = \frac{1}{t}\int_0^t dt'\left(\sqrt{\frac{\kappa}{2}}\,\langle\phi(t')|\,X_c\,|\phi(t')\rangle + \frac{1}{2}\frac{d\zeta_s(t')}{dt'}\right)
\tag{D.10}
$$

612 Remarkably, despite the presence of the tensor term, we can relate the signal to $\bar{p}$ using the
613 expressions of the wave function in the truncated basis (27) and (28) and we find:

$$\sigma(t) = \frac{1}{t} \int_0^t dt' \left( \sqrt{\Gamma}\,\bar{p}(t') + \frac{1}{2}\frac{d\zeta_s(t')}{dt'} \right) \tag{D.11}$$

614 We now want to access the mean and variance of $P$ conditioned to the value $\mathcal{S}$ of the homodyne
615 signal $\sigma$. It can be shown [23] that the conditional mean is always proportional to the signal,
616 and that the conditional variance, synonymous with metrological gain, depends on time but
617 not on the signal:

$$\langle P \rangle_{\sigma=\mathcal{S}} = m(t)\frac{\mathcal{S}}{\sqrt{\Gamma}} \quad \text{with} \quad m(t) = \sqrt{\Gamma}\,\frac{\langle \sigma(t)\bar{p}(t) \rangle_{\text{stoch}}}{\langle \sigma^2(t) \rangle_{\text{stoch}}} \tag{D.12}$$

618

$$\text{Var}_{\sigma=\mathcal{S}}(P) = \mathcal{V}(t) \quad \text{with} \quad \mathcal{V}(t) = \frac{1}{4u(t)} + \langle \bar{p}^2(t) \rangle_{\text{stoch}} - \frac{\langle \sigma(t)\bar{p}(t) \rangle_{\text{stoch}}^2}{\langle \sigma^2(t) \rangle_{\text{stoch}}} \tag{D.13}$$

619 where $\langle \cdots \rangle_{\text{stoch}}$ at time $t$ indicates that the mean is taken over all realizations of stochastic
620 processes $d\zeta_s(t')$ and $d\zeta_a(t')$ over time interval $[0, t]$. Furthermore, the sum of the first two
621 terms of the conditional variance corresponds exactly to $\langle P^2 \rangle$, which was calculated in section
622 3. Indeed, with this formalism, we can calculate the variance of the operator $P$, which has a
623 zero mean $\langle P \rangle \equiv \langle \bar{p} \rangle_{\text{stoch}} = 0$:

$$
\begin{aligned}
\langle P^2 \rangle(\tau) &\equiv \langle \langle \phi(\tau)| P^2 |\phi(\tau) \rangle \rangle_{\text{stoch}} \\
&= \langle \langle \phi(\tau)| P^2 |\phi(\tau) \rangle - \langle \phi(\tau)| P |\phi(\tau) \rangle^2 + \langle \phi(\tau)| P |\phi(\tau) \rangle^2 \rangle_{\text{stoch}} \\
&= \langle \text{Var}_{\phi(\tau)} P \rangle_{\text{stoch}} + \langle \bar{p}^2(\tau) \rangle_{\text{stoch}} \\
&= \frac{1}{4u(\tau)} + \langle \bar{p}^2(\tau) \rangle_{\text{stoch}} \\
&= \frac{1}{4u(\tau)} + \int_0^\tau d\tau' e^{-2(\epsilon+\tilde{\gamma}/2)(\tau-\tau')}\left[ \left(\frac{1}{2u(\tau')}-\epsilon\right)^2 + \frac{\tilde{\gamma}}{2}\left(1-\frac{1}{2u(\tau')}\right)^2 \right] \\
&= (23)
\end{aligned}
\tag{D.14}
$$

624 Equations (D.9) and (D.11) allow us to calculate the variance and covariance by averaging
625 these two stochastic processes. By introducing the Langevin forces $\frac{d\zeta(\tau)}{d\tau}$ into the integrals and
626 using the fact that, when switching to the stochastic mean $\langle \frac{d\zeta(\tau)}{d\tau}\frac{d\zeta(\tau')}{d\tau'} \rangle_{\text{stoch}} = \delta(\tau-\tau')$, we
627 obtain:

$$
\begin{aligned}
\frac{\langle \sigma\,\bar{p} \rangle_{\text{stoch}}}{\sqrt{\Gamma}} &= \frac{1}{\tau}\int_0^\tau d\tau' e^{-(\epsilon+\frac{\tilde{\gamma}}{2})(\tau-\tau')}\left\{ \left(\left(\frac{1}{2u(\tau')}-\epsilon\right)^2 + \frac{\tilde{\gamma}}{2}\left(1-\frac{1}{2u(\tau')}\right)^2\right)\frac{1-e^{-(\epsilon+\frac{\tilde{\gamma}}{2})(\tau-\tau')}}{\epsilon+\tilde{\gamma}/2} + \frac{1}{2}\left(\frac{1}{2u(\tau')}-\epsilon\right)\right\} \\
&= \frac{1-\epsilon}{2\tau}\frac{1-e^{-(\epsilon+\frac{\tilde{\gamma}}{2})\tau}}{(\epsilon+\tilde{\gamma}/2)^2}\left(\epsilon\,e^{-(\epsilon+\frac{\tilde{\gamma}}{2})\tau}+\tilde{\gamma}/2\right)
\end{aligned}
\tag{D.15}
$$

$$
\begin{aligned}
\frac{\langle \sigma^2 \rangle_{\text{stoch}}}{\Gamma} &= \frac{1}{\tau^2}\int_0^\tau d\tau'\left\{ \left(\frac{1}{2}+\left(\frac{1}{2u(\tau')}-\epsilon\right)\frac{1-e^{-(\epsilon+\frac{\tilde{\gamma}}{2})(\tau-\tau')}}{\epsilon+\tilde{\gamma}/2}\right)^2 + \frac{\tilde{\gamma}}{2}\left(1-\frac{1}{2u(\tau')}\right)^2\left(\frac{1-e^{-(\epsilon+\frac{\tilde{\gamma}}{2})(\tau-\tau')}}{\epsilon+\tilde{\gamma}/2}\right)^2\right\} \\
&= \frac{1}{(2\epsilon+\tilde{\gamma})^3\tau^2}\left[ 4(1-\epsilon)(2\epsilon-\tilde{\gamma})[1-e^{-(2\epsilon+\tilde{\gamma})\tau/2}] - 4\epsilon(1-\epsilon)[1-e^{-(2\epsilon+\tilde{\gamma})\tau}] \right. \\
&\quad \left. + \left[(2\epsilon+\tilde{\gamma})(2\tilde{\gamma}+(\epsilon-\tilde{\gamma}/2)^2)\right]\tau \right]
\end{aligned}
\tag{D.16}
$$

628   By inserting expressions (D.15) and (D.16) into (D.12) and (D.13), we calculate exactly
629   the mean and variance conditioned to the integrated signal $\sigma$. The coefficient $m(\tau)$ has a
630   maximum that we calculate for $\epsilon, \tilde{\gamma} \ll 1$ and $\tau \gg 1$. More precisely, we renormalize $\tau$ and $\tilde{\gamma}$:

$$\bar{\tau} \equiv \sqrt{\epsilon}\tau \; ; \; \bar{\gamma} \equiv \frac{\tilde{\gamma}}{\epsilon} \tag{D.17}$$

631   and we take the limit $\epsilon \to 0$ at $\bar{\tau}, \bar{\gamma}$ fixed. We obtain:

$$m(\bar{\tau}) = 1 - \frac{\sqrt{\epsilon}}{2}\left(\frac{1}{\bar{\tau}} + \left(1 + \frac{\bar{\gamma}}{6}\right)\bar{\tau}\right)$$

$$\mathcal{V}(\bar{\tau}) = \sqrt{\epsilon}\left(\frac{1}{4\bar{\tau}} + \frac{\bar{\gamma}}{6}\bar{\tau}\right) \tag{D.18}$$

632   We therefore have the following expressions for the maximum quasi-QND squeezing time, the
633   value of the conditional mean and variance associated with it:

$$\tau_{\text{QND}}^{\max} \simeq \frac{1}{\sqrt{\epsilon + \frac{\tilde{\gamma}}{6}}} \underset{\tilde{\gamma} \ll \epsilon}{\simeq} \frac{1}{\sqrt{\epsilon}} \; ; \; m_{\text{QND}} \simeq 1 - \sqrt{\epsilon + \frac{\tilde{\gamma}}{6}} \underset{\tilde{\gamma} \ll \epsilon}{\simeq} 1 - \sqrt{\epsilon} \; ; \; \mathcal{V}_{\text{QND}} \simeq \frac{1}{4}\sqrt{\epsilon + \tilde{\gamma}/6}\left(1 + \frac{2}{3}\frac{\tilde{\gamma}}{\epsilon + \tilde{\gamma}/6}\right) \underset{\tilde{\gamma} \ll \epsilon}{\simeq} \frac{\sqrt{\epsilon}}{4}$$

$$\tag{D.19}$$

# E   3-mode stationary solutions for $^3$He

635   In this section, we give the exact stationary solutions of systems (52) and (53). The solution
636   of each system is expressed in terms of $\langle X_\beta P_c \rangle$ and $\langle P_\beta X_c \rangle$, respectively.

$$
\begin{cases}
\langle X_\alpha P_c \rangle = 0 \\[4pt]
\langle X_\beta^2 \rangle = \frac{1}{2} + \frac{2\Omega_{V\beta}}{\gamma_\beta}\langle X_\beta P_c \rangle \\[4pt]
\langle X_\alpha^2 \rangle = \frac{1}{2\epsilon} - 2\Omega_{V_\beta}\left(\frac{1}{\kappa} + \frac{1}{\gamma_\beta}\right)\langle X_\beta P_c \rangle \\[4pt]
\langle X_\beta P_c \rangle\left[1 + \frac{4\Omega_{V_\alpha}\Omega_{T_\alpha}}{\gamma_\beta(\gamma_\beta + \kappa)} + \frac{4\Omega_{V_\beta}\Omega_{T_\beta}}{\gamma_\beta(\gamma_\beta + \kappa)} + \frac{4\Omega_{V_\beta}\Omega_{T_\beta}}{\kappa(\gamma_\beta + \kappa)}\right] = (1 - \epsilon)\frac{\Omega_{V_\beta}}{\gamma_\beta + \kappa} \\[4pt]
\langle X_\alpha X_\beta \rangle = \frac{2\Omega_{V\alpha}}{\gamma_\beta}\langle X_\beta P_c \rangle \\[4pt]
\langle P_c^2 \rangle = \frac{1}{2} - \frac{2\Omega_{T\beta}}{\kappa}\langle X_\beta P_c \rangle
\end{cases}
\tag{E.1}
$$

$$
\begin{cases}
\langle P_\alpha X_c \rangle = 0 \\[4pt]
\langle P_\beta^2 \rangle = \frac{1}{2} - \frac{2\Omega_{T\beta}}{\gamma_\beta}\langle P_\beta X_c \rangle \\[4pt]
\langle P_\alpha^2 \rangle = \frac{\epsilon}{2} + 2\epsilon\Omega_{V_\beta}\left(\frac{1}{\kappa} + \frac{1}{\gamma_\beta}\right)\langle P_\beta X_c \rangle \\[4pt]
\langle P_\beta X_c \rangle\left[1 + \frac{4\Omega_{V_\alpha}\Omega_{T_\alpha}}{\gamma_\beta(\gamma_\beta + \kappa)} + \frac{4\Omega_{V_\beta}\Omega_{T_\beta}}{\gamma_\beta(\gamma_\beta + \kappa)} + \frac{4\Omega_{V_\beta}\Omega_{T_\beta}}{\kappa(\gamma_\beta + \kappa)}\right] = (1 - \epsilon)\frac{\Omega_{V_\beta}}{\gamma_\beta + \kappa} \\[4pt]
\langle P_\alpha P_\beta \rangle = -\frac{2\Omega_{T\alpha}}{\gamma_\beta}\langle P_\beta X_c \rangle \\[4pt]
\langle X_c^2 \rangle = \frac{1}{2} + \frac{2\Omega_{V\beta}}{\kappa}\langle P_\beta X_c \rangle
\end{cases}
\tag{E.2}
$$

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
