# Peer review of "Deterministic nuclear spin squeezing and squeezing by continuous measurement using vector and tensor light shifts"

_SciPost Physics_

## Round 1 · Referee Report · Anonymous (Referee 1) · 2025-9-27

Report

The paper is about the evaluation of how the tensor component of the interaction Hamiltonian of nuclear spins (alkali atoms) and a detuned light field affects the squeezing of the spins. The authors identify two regimes of squeezing, a quasi-QND and a deterministic one. They consider the so-called Holstein-Primakoff approximation to obtain a model Hamiltonian for the system. Then, they solve a two-mode master equation where they also take into account decoherence for the atomic mode and photon loss in the cavity. From this they obtain the expectation values of the quadrature of the atomic mode. They apply these results to ytterbium 173, strontium 87, and also, He 3. In doing so, the authors consider cavities from real experiments. As far as I understood, the main result is that considering the tensor term, “better squeezing” can be achieved in the deterministic regime. I believe the results are important as due to their long coherence times, these atoms are frequently used in quantum metrology and quantum information processing.

The paper is well-written; the formulas and derivations seem correct. It is rather technical for a reader without sufficient background, but the provided appendices and referenced literature definitely help with grasping the main ideas and derivations. In my opinion, the paper has the potential to be useful for future squeezing experiments with alkaline earth atoms.

Therefore, I recommend its publication in SciPost Physics after the authors do some minor revisions on the manuscript so that it is more accessible to an even broader audience.

I have the following comments, questions:
• The authors could also mention entanglement detection as an application of squeezing in the Introduction part.
• Can the authors explain briefly somewhere (maybe already in the Introduction) what is the main difference between the two squeezing regimes (quasi-QND and deterministic) intuitively? What advantage has one compared to the other? And why is this missing in the f=1/2 case?
• Also, is there a simple, intuitive explanation why does the tensor term introduce this new regime or one just has to look at Section 3.1 and understand through the formulas?
• The authors say that „In experimental works, those terms [tensor type terms] are either neglected for a large detuning, or else neutralized by dynamic decoupling.” Why is this neutralized if it can in principle provide advantage?
• In the Introduction it would be good to briefly mention the main results beyond “For each regime, we quantify analytically the metrological gain as a function of the atomic parameters, including decoherence.” Especially, what is the advantage of considering the tensor part of the interaction?
• It would be nice to have a brief description of the subsequent Sections’ content at the end of the Introduction.
• I would explicitly state near Eq. (1) that Appendix A explains where does Eq. (1) come from to help the reader.
• Would it be possible to change the first plots (for \epsilon) for Yb and Sr (in Fig 3 and Fig 4) so that they are both logarithmic or non-logarithmic? If it is possible, in my opinion it would look better if these plots were unified.
• Are there experiments where they did squeezing without the tensor term? It would be great to see a table (if there are enough experiments) with how the squeezing can be improved by considering the tensor terms in „real life” situations.
• In Section 3.2. what is needed for the alpha^t to cancel? (only the detuning is needed?) It would be nice to remind the reader of the physical content of these parameters from time to time. This goes for other parameters as well, as there are many of them (for example in Figs 3-5).
• In Section 3.2, is it true that the higher the detuning is, the better squeezing we have? Are there drawbacks of having a very high detuning? It would be nice to have some comments on this.
• Many equations have „avec” in them (e.g. Eq. (32)). It would be better to write it in English.
• Regarding helium 3 the authors say that the previous theory holds with minor modifications. Can the authors elaborate what are these minor modifications?
• What is the main conclusion of the paper? Is it that a better squeezing can be achieved with considering the tensor part? It should be emphasized more both in the Introduction and Conclusion.

Recommendation

Ask for minor revision

---

## Round 1 · Referee Report · Anonymous (Referee 2) · 2025-11-18

Disclosure of Generative AI use

The referee discloses that the following generative AI tools have been used in the preparation of this report:

AI was used as writing assistant for checking grammar and spelling.

Strengths

1. The study addresses quantum control of nuclear spin through continuous measurement, which remains a timely and actively developing direction in atomic and quantum-optical research.
2. It extends established squeezing methods to Strontium, Ytterbium, and Helium, offering an application scope that, to my knowledge, has not been explored in previous literature.
3. The presentation is largely self-contained, providing detailed derivations that guide the reader through the theoretical framework without relying heavily on external references.
4. The discussion is supported by an extensive set of figures that illustrate the relevant regimes, dynamical behavior, and parameter dependencies in a clear and systematic manner.

Weaknesses

1. The effective Hamiltonian of the form P P_c + \epsilon X X_c, arising from second-rank tensor polarizability, is presented as a central result, but similar non-QND couplings have been identified previously in the context of atomic ensembles probed via Faraday rotation, even if not specifically linked to nuclear-spin squeezing.
2. The dynamics governed by the stochastic Schrödinger or master equation for Gaussian states, including evolution of means and variances, has been treated in earlier work; employing this established formalism directly could strengthen the analysis of the present special case.

Report

Overall, the manuscript meets the journal’s acceptance criteria, and the results are well aligned with the scope of the venue. I consider the paper suitable for publication, and I agree with the authors’ statement that the work opens a new pathway within an active research direction, with clear potential for substantial follow-up studies. The additional claim that it constitutes a breakthrough on a long-standing obstacle appears somewhat strong; in my view the contribution represents a significant and well-motivated advance rather than a decisive resolution of a prior bottleneck.

Requested changes

Intro: 1) "Recently, another theoretical method of spin squeezing..." what is a theoretical method? Add more context of this method and compare.. 2) "In experimental work, those terms [2nd rank tensor polarisability] are either neglected ... or neutralise...". This is not quite true. Please consults the works in the lab of Eugene Polzik e.g. on stationary entanglement of atomic spins [Hanah Krauter et al.] a related theory work by Christine Muschik and Ignacio Cirac. this work shows the derivation of the PPc+epsilon*XXc structure of the effective Hamiltonian. I ask the authors to relate their work to this previous papers and adapt claims where necessary.

Sec 2: 3) 2.1 lacks reference to Appendix A 4) Eq. (3) assumes a dilute (maybe still optically thick) ensemble. 5) The notation changes from a_alpha to a_k without note. 6) Eq. (13): A similar Hamiltonian has been derived by Muschik et al., see remark above. 7) In Eq. (18) the Lindblad term for atoms would rather correspond to optical pumping, not to spontaneous emission. Maybe the authors can comment on this, and also on the necessity to maintain large polarisation in steady state. 8) The parameter epsilon should be introduced more clearly, along with its domain. This becomes clear only later and in footnote 2.

Sec 3: 9) Fig 2: Why a.u. on the y-axis? The zoom-in appears inconsistent with the figure.

Sec 4: 10) I would recommend to use the general framework for solving the stochastic master equation for Gaussian dynamics in terms of displacement vector and covariance matrix. It is a general feature that the covariance matrix evolves deterministically in this case.

Sec 5: 11) Consider writing a Hamiltonian instead of the matrix equation (43).

Recommendation

Publish (easily meets expectations and criteria for this Journal; among top 50%)

---

## Round 2 · Referee Report · Anonymous (Referee 1) · 2025-12-20

Report

The authors have addressed properly all my comments, therefore I suggest the publication of their work.

Recommendation

Publish (meets expectations and criteria for this Journal)

---

## Round 2 · Author Response

Dear Editor,
We would like to thank you and the two reviewers for the quality and insightfulness of the reports we received.
Based on their comments, we have made several changes (marked in red in a file merged with the detailed answers to the referees) which we believe have improved the clarity of the article.
Sincerely yours,
The authors

---

## Editorial Decision

in_refereeing